# Policy Regret in Repeated Games

**Raman Arora**
Dept. of Computer Science
Johns Hopkins University
Baltimore, MD 21204
arora@cs.jhu.edu

**Michael Dinitz**
Dept. of Computer Science
Johns Hopkins University
Baltimore, MD 21204
mdinitz@cs.jhu.edu

**Teodor V. Marinov**
Dept. of Computer Science
Johns Hopkins University
Baltimore, MD 21204
tmarino2@jhu.edu

**Mehryar Mohri**
Courant Institute and Google Research
New York, NY 10012
mohri@cims.nyu.edu

## Abstract

The notion of *policy regret* in online learning is a well defined performance measure for the common scenario of adaptive adversaries, which more traditional quantities such as external regret do not take into account. We revisit the notion of policy regret and first show that there are online learning settings in which policy regret and external regret are incompatible: any sequence of play that achieves a favorable regret with respect to one definition must do poorly with respect to the other. We then focus on the game-theoretic setting where the adversary is a self-interested agent. In that setting, we show that external regret and policy regret are not in conflict and, in fact, that a wide class of algorithms can ensure a favorable regret with respect to both definitions, so long as the adversary is also using such an algorithm. We also show that the sequence of play of no-policy regret algorithms converges to a *policy equilibrium*, a new notion of equilibrium that we introduce. Relating this back to external regret, we show that coarse correlated equilibria, which no-external regret players converge to, are a strict subset of policy equilibria. Thus, in game-theoretic settings, every sequence of play with no external regret also admits no policy regret, but the converse does not hold.

## 1  Introduction

Learning in dynamically evolving environments can be described as a repeated game between a player, an online learning algorithm, and an adversary. At each round of the game, the player selects an action, e.g. invests in a specific stock, the adversary, which may be the stock market, chooses a utility function, and the player gains the utility value of its action. The player observes the utility value and uses it to update its strategy for subsequent rounds. The player's goal is to accumulate the largest possible utility over a finite number of rounds of play.[1]

The standard measure of the performance of a player is its *regret*, that is the difference between the utility achieved by the best offline solution from some restricted class and the utility obtained by the online player, when utilities are revealed incrementally. Formally, we can model learning as the following problem. Consider an action set $\mathcal{A}$. The player selects an action $a_t$ at round $t$, the adversary picks a utility function $u_t$, and the player gains the utility value $u_t(a_t)$. While in a full observation setting the player observes the entire utility function $u_t$, in a *bandit setting* the player only observes

the utility value of its own action, $u_t(a_t)$. We use the shorthand $a_{1:t}$ to denote the player's sequence of actions $(a_1, \ldots, a_t)$ and denote by $\mathcal{U}_t = \{u : \mathcal{A}^t \to \mathbb{R}\}$ the family of utility functions $u_t$. The objective of the player is to maximize its expected cumulative utility over $T$ rounds, i.e. maximize $\mathbb{E}[\sum_{t=1}^{T} f_t(a_{1:t})]$, where the expectation is over the player's (possible) internal randomization. Since this is clearly impossible to maximize without knowledge of the future, the algorithm instead seeks to achieve a performance comparable to that of the best fixed action in hindsight. Formally, *external regret* is defined as

$$R(T) = \mathbb{E}\left[\max_{a \in \mathcal{A}} \sum_{t=1}^{T} u_t(a_{1:t-1}, a) - \sum_{t=1}^{T} u_t(a_{1:t})\right]. \tag{1}$$

A player is said to *admit no external regret* if the external regret is sublinear, that is $R(T) = o(T)$. In contrast to statistical learning, online learning algorithms do not need to make stochastic assumptions about data generation: strong regret bounds are possible even if the utility functions are adversarial.

There are two main adversarial settings in online learning: the *oblivious setting* where the adversary ignores the player's actions and where the utility functions can be thought of as determined before the game starts (for instance, in weather prediction); and the *adaptive setting* where the adversary can react to the player's actions, thus seeking to throw the player off track (e.g., competing with other agents in the stock market). More generally, we define an $m$-*memory bounded adversary* as one that at any time $t$ selects a utility function based on the player's past $m$ actions: $u_t(a'_1, \ldots, a'_{t-m-1}, a_{t-m}, \ldots, a_t) = u_t(a_1, \ldots, a_{t-m-1}, a_{t-m}, \ldots, a_t)$, for all $a'_1, \ldots, a'_{t-m-1}$ and all $a_1, \ldots, a_t$. An oblivious adversary can therefore be equivalently viewed as a 0-memory bounded adversary and an adaptive adversary as an $\infty$-memory bounded adversary. For an oblivious adversary, external regret in Equation 1 reduces to $R(T) = \mathbb{E}[\max_{a \in \mathcal{A}} \sum_{t=1}^{T} u_t(a) - u_t(a_t)]$, since the utility functions do not depend upon past actions. Thus, external regret is meaningful when the adversary is oblivious, but it does not admit any natural interpretation when the adversary is adaptive. The problem stems from the fact that in the definition of external regret, the benchmark is a function of the player's actions. Thus, if the adversary is adaptive, or even memory-bounded for some $m > 0$, then, external regret does not take into account how the adversary would react had the player selected some other action.

To resolve this critical issue, Arora et al. (2012b) introduced an alternative measure of performance called *policy regret* for which the benchmark does not depend on the player's actions. Policy regret is defined as follows

$$P(T) = \max_{a \in \mathcal{A}} \sum_{t=1}^{T} u_t(a, \ldots, a) - \mathbb{E}\left[\sum_{t=1}^{T} u_t(a_{1:t})\right]. \tag{2}$$

Arora et al. (2012b) further gave a reduction, using a mini-batch technique where the minibatch size is larger than the memory $m$ of adversary, that turns any algorithm with a sublinear external regret against an oblivious adversary into an algorithm with a sublinear policy regret against an $m$-memory bounded adversary, albeit at the price of a somewhat worse regret bound, which is still sublinear.

In this paper, we revisit the problem of online learning against adaptive adversaries. Since Arora et al. (2012b) showed that there exists an adaptive adversary against which any online learning algorithm admits linear policy regret, even when the external regret may be sublinear, we ask if no policy regret implies no external regret. One could expect this to be the case since policy regret seems to be a *stronger* notion than external regret. However, our first main result (Theorem 3.2) shows that this in fact is *not* the case and that the two notions of regret are incompatible: there exist adversaries (or sequence of utilities) on which action sequences with sublinear external regret admit linear policy regret and action sequences with sublinear policy regret incur linear external regret.

We argue, however, that such sequences may not arise in practical settings, that is in settings where the adversary is a self-interested entity. In such settings, rather than considering a malicious opponent whose goal is to hurt the player by inflicting large regret, it seems more reasonable to consider an opponent whose goal is to maximize his own utility. In zero-sum games, maximizing one's utility comes at the expense of the other player's, but there is a subtle difference between an adversary who is seeking to maximize the player's regret and an adversary who is seeking to minimize the player's utility (or maximize his own utility). We show that in such strategic game settings there is indeed a strong relationship between external regret and policy regret. In particular, we show in Theorem 3.4 that a large class of *stable* online learning algorithms with sublinear external regret also benefit from sublinear policy regret.

Further, we consider a two-player game where each player is playing a no policy regret algorithm. It is known that no external regret play converges to a coarse correlated equilibrium (CCE) in such a game, but what happens when players are using no policy regret algorithms? We show in Theorem 4.8 that the average play in repeated games between no policy regret players converges to a *policy equilibrium*, a new notion of equilibrium that we introduce. Policy equilibria differ from more traditional notions of equilibria such as Nash or CCEs in a crucial way. Recall that a CCE is defined to be a recommended joint strategy for players in a game such that there is no incentive for any player to deviate unilaterally from the recommended strategy if other players do not deviate.

What happens if the other players react to one player's deviation by deviating themselves? This type of reasoning is not captured by external regret, but is essentially what is captured by policy regret. Thus, our notion of policy equilibrium must take into account these counterfactuals, and so the definition is significantly more complex. But, by considering functions rather than just actions, we can define such equilibria and prove that they exactly characterize no policy regret play.

Finally, it becomes natural to determine the relationship between policy equilibria (which characterize no policy regret play) and CCEs (which characterize no external regret play). We show in Theorems 4.9 and 4.10 that the set of CCEs is a strict subset of policy equilibria. In other words, every CCE can be thought of as a policy regret equilibrium, but no policy regret play might not converge to a CCE.

# 2    Related work

The problem of minimizing policy regret in a fully adversarial setting was first studied by Merhav et al. (2002). Their work dealt specifically with the full observation setting and assumed that the utility (or loss) functions were $m$-memory bounded. They gave regret bounds in $O(T^{2/3})$. The follow-up work by Farias and Megiddo (2006) designed algorithms in a reactive bandit setting. However, their results were not in the form of regret bounds but rather introduced a new way to compare against acting according to a fixed expert strategy. Arora et al. (2012b) studied $m$-memory bounded adversaries both in the bandit and full information settings and provided extensions to more powerful competitor classes considered in swap regret and more general $\Phi$-regret. Dekel et al. (2014) provided a lower bound in the bandit setting for switching cost adversaries, which also leads to a tight lower bound for policy regret in the order $\Omega(T^{2/3})$. Their results were later extended by Koren et al. (2017a) and Koren et al. (2017b). More recently, Heidari et al. (2016) considered the multi-armed bandit problem where each arm's loss evolves with consecutive pulls. The process according to which the loss evolves was not assumed to be stochastic but it was not arbitrary either – in particular, the authors required either the losses to be concave, increasing and to satisfy a decreasing marginal returns property, or decreasing. The regret bounds given are in terms of the time required to distinguish the optimal arm from all others.

A large part of reinforcement learning is also aimed at studying sequential decision making problems. In particular, one can define a Markov Decision Process (MDP) by a set of states equipped with transition distributions, a set of actions and a set of reward or loss distributions associated with each state action pair. The transition and reward distributions are assumed unknown and the goal is to play according to a strategy that minimizes loss or maximizes reward. We refer the reader to (Sutton and Barto, 1998; Kakade et al., 2003; Szepesvári, 2010) for general results in RL. MDPs in the online setting with bandit feedback or arbitrary payoff processes have been studied by Even-Dar et al. (2009a); Yu et al. (2009); Neu et al. (2010) and Arora et al. (2012a).

The tight connection between no-regret algorithms and correlated equilibria was established and studied by Foster and Vohra (1997); Fudenberg and Levine (1999); Hart and Mas-Colell (2000); Blum and Mansour (2007). A general extension to games with compact, convex strategy sets was given by Stoltz and Lugosi (2007). No external regret dynamics were studied in the context of socially concave games (Even-Dar et al., 2009b). More recently, Hazan and Kale (2008) considered more general notions of regret and established an equivalence result between fixed-point computation, the existence of certain no-regret algorithms, and the convergence to the corresponding equilibria. In a follow-up work by Mohri and Yang (2014) and Mohri and Yang (2017), the authors considered a more powerful set of competitors and showed that the repeated play according to conditional swap-regret or transductive regret algorithms leads to a new set of equilibria.

# 3   Policy regret in reactive versus strategic environments

Often, distant actions in the past influence an adversary more than more recent ones. The definition of policy regret (2) models this influence decay by assuming that the adversary is $m$-memory bounded for some $m \in \mathbb{N}$. This assumption is somewhat stringent, however, since ideally we could model the current move of the adversary as a function of the entire past, even if actions taken further in the past have less significance. Thus, we extend the definition of Arora et al. (2012b) as follows.

**Definition 3.1.**   The $m$-*memory policy regret* at time $T$ of a sequence of actions $(a_t)_{t=1}^T$ with respect to a fixed action $a$ in the action set $\mathcal{A}$ and the sequence of utilities $(u_t)_{t=1}^T$, where $u_t : \mathcal{A}^t \to \mathbb{R}$ and $m \in \mathbb{N} \bigcup \{\infty\}$ is

$$P(T, a) = \sum_{t=1}^T u_t(a_1, \cdots, a_{t-m}, a, \cdots, a) - \sum_{t=1}^T u_t(a_1, \cdots, a_t).$$

We say that the sequence $(a_t)_{t=1}^T$ has sublinear policy regret (or no policy regret) if $P(T, a) < o(T)$, for all actions $a \in \mathcal{A}$.

Let us emphasize that this definition is just an extension of the standard policy regret definition and that, when the utility functions are $m$-memory bounded, the two definitions exactly coincide.

While the motivation for policy regret suggests that this should be a stronger notion compared to external regret, we show that not only that these notions are incomparable in the general adversarial setting, but that they are also *incompatible* in a strong sense.

**Theorem 3.2.**   *There exists a sequence of $m$-memory bounded utility functions $(u_t)_{t=1}^T$, where $u_t : \mathcal{A} \to \mathbb{R}$, such that for any constant $m \geq 2$ (independent of $T$), any action sequence with sublinear policy regret will have linear external regret and any action sequence with sublinear external regret will have linear policy regret.*

The proof of the above theorem constructs a sequence for which no reasonable play can attain sublinear external regret. In particular, the only way the learner can have sublinear external regret is if they choose to have very small utility. To achieve this, the utility functions chosen by the adversary are the following. At time $t$, if the player chose to play the same action as their past 2 actions then they get utility $\frac{1}{2}$. If the player's past two actions were equal but their current action is different, then they get utility $1$, and if their past two actions differ then no matter what their current action is they receive utility $0$. It is easy to see that the maximum utility play for this sequence (and the lowest 2-memory bounded policy regret strategy) is choosing the same action at every round. However, such an action sequence admits linear external regret. Moreover, every sublinear external regret strategy must then admit sublinear utility and thus linear policy regret.

As discussed in Section 1, in many realistic environments we can instead think of the adversary as a self-interested agent trying to maximize their own utility, rather than trying to maximize the regret of the player. This more strategic environment is better captured by the game theory setting, in particular a 2-player game where both players are trying to maximize their utility. Even though we have argued that external regret is not a good measure, our next result shows that minimizing policy regret in games can be done if both players choose their strategies according to certain no external regret algorithms. More generally, we adapt a classical notion of stability from the statistical machine learning setting and argue that if the players use no external regret algorithms that are *stable*, then the players will have no policy regret in expectation. To state the result formally we first need to introduce some notation.

**Game definition:**   We consider a 2-player game $\mathcal{G}$, with players 1 and 2. The action set of player $i$ is denoted by $\mathcal{A}_i$, which we think of as being embedded into $\mathbb{R}^{|\mathcal{A}_i|}$ in the obvious way where each action corresponds to a standard basis vector. The corresponding simplex is $\Delta \mathcal{A}_i$. The action of player 1 at time $t$ is $a_t$ and of player 2 is $b_t$. The observed utility for player $i$ at time $t$ is $u_i(a_t, b_t)$ and this is a bi-linear form with corresponding matrix $\mathrm{P}_i$. We assume that the utilities are bounded in $[0, 1]$.

**Algorithm of the player:**   When discussing algorithms, we take the view of player 1. Specifically, at time $t$, player 1 plays according to an algorithm which can be described as $Alg_t : (\mathcal{A}_1 \times \mathcal{A}_2)^t \to \Delta \mathcal{A}_1$. We distinguish between two settings: full information, in which the player observes the full utility

function at time $t$ (i.e., $u_1(\cdot, b_t)$), and the bandit setting, in which the player only observes $u_1(a_t, b_t)$. In the full information setting, algorithms like multiplicative weight updates (MWU Arora et al. (2012c)) depend only on the past $t-1$ utility functions $(u_1(\cdot, b_\ell))_{\ell=1}^{t-1}$, and thus we can think of $Alg_t$ as a function $f_t : \mathcal{A}_2^t \to \Delta\mathcal{A}_1$. In the bandit setting, though, the output at time $t$ of the algorithm depends both on the previous $t-1$ actions $(a_\ell)_{\ell=1}^{t-1}$ and on the utility functions (i.e., the actions picked by the other player).

But even in the bandit setting, we would like to think of the player's algorithm as a function $f_t : \mathcal{A}_2^t \to \Delta\mathcal{A}_1$. We cannot quite do this, however we *can* think of the player's algorithm as a *distribution* over such functions. So how do we remove the dependence on $\mathcal{A}_1^t$? Intuitively, if we fix the sequence of actions played by player 2, we want to take the expectation of $Alg_t$ over possible choices of the $t$ actions played by player 1. In order to do this more formally, consider the distribution $\mu$ over $\mathcal{A}_1^{t-1} \times \mathcal{A}_2^{t-1}$ generated by simulating the play of the players for $t$ rounds. Then let $\mu_{b_{0:t}}$ be the distribution obtained by conditioning $\mu$ on the actions of player 2 being $b_{0:t}$. Now we let $f_t(b_{0:t-1})$ be the distribution obtained by sampling $a_{0:t-1}$ from $\mu_{b_{1:t-1}}$ and using $Alg(a_{0:t-1}, b_{0:t-1})$. When taking expectations over $f_t$, the expectation is taken with respect to the above distribution. We also refer to the output $p_t = f_t(b_{0:t-1})$ as the strategy of the player at time $t$.

Now that we can refer to algorithms simply as functions (or distributions over functions), we introduce the notion of a stable algorithm.

**Definition 3.3.** Let $f_t : \mathcal{A}_2^t \to \Delta\mathcal{A}_1$ be a sample from $Alg_t$ (as described above), mapping the past $t$ actions in $\mathcal{A}_2$ to a distribution over the action set $\mathcal{A}_1$. Let the distribution returned at time $t$ be $p_t^1 = f_t(b_1, \ldots, b_t)$. We call this algorithm *on average* $(m, S(T))$ *stable* with respect to the norm $\|\cdot\|$, if for any $b'_{t-m+1}, \ldots, b'_t \in \mathcal{A}_2$ such that $\tilde{p}_t^1 = f_t(b_1, \ldots, b_{t-m}, b'_{t-m+1}, \ldots, b'_t) \in \Delta\mathcal{A}_1$, it holds that $\mathbb{E}[\sum_{t=1}^T \|p_t^1 - \tilde{p}_t^1\|] \le S(T)$, where the expectation is taken with respect to the randomization in the algorithm.

Even though this definition of stability is given with respect to the game setting, it is not hard to see that it can be extended to the general online learning setting, and in fact this definition is similar in spirit to the one given in Saha et al. (2012). It turns out that most natural no external regret algorithms are stable. In particular we show, in the supplementary, that both Exp3 Auer et al. (2002) and MWU are on average $(m, m\sqrt{T})$ stable with respect to $\ell_1$ norm for any $m < o(\sqrt{T})$. It is now possible to show that if each of the players are facing stable no external regret algorithms, they will also have bounded policy regret (so the incompatibility from Theorem 3.2 cannot occur in this case).

**Theorem 3.4.** *Let* $(a_t)_{t=1}^T$ *and* $(b_t)_{t=1}^T$ *be the action sequences of player* 1 *and* 2 *and suppose that they are coming from no external regret algorithms modeled by functions* $f_t$ *and* $g_t$, *with regrets* $R_1(T)$ *and* $R_2(T)$ *respectively. Assume that the algorithms are on average* $(m, S(T))$ *stable with respect to the* $\ell_2$ *norm. Then*

$$\mathbb{E}\left[P(T, a)\right] \le \|\mathrm{P}_1\| S(T) + R_1(T)$$
$$\mathbb{E}\left[P(T, b)\right] \le \|\mathrm{P}_2\| S(T) + R_2(T),$$

*where* $u_t(a_{1:t})$ *in the definition of* $P(T, a)$ *equals* $u_1(a_t, g_t(a_{0:t-1}))$ *and similarly in the definition of* $P(T, b)$, *equals* $u_2(b_t, f_t(b_{0:t-1}))$. *The above holds for any fixed actions* $b \in \mathcal{A}_2$ *and* $a \in \mathcal{A}_1$. *Here the matrix norm* $\|\cdot\|$ *is the spectral norm.*

## 4  Policy equilibrium

Recall that unlike external regret, policy regret captures how other players in a game might react if a player decides to deviate from their strategy. The story is similar when considering different notions of equilibria. In particular Nash equlibria, Correlated equilibria and CCEs can be interpreted in the following way: if player $i$ deviates from the equilibrium play, their utility will not increase no matter how they decide to switch, provided that *all other players continue to play according to the equilibrium*. This sentiment is a reflection of what no external and no swap regret algorithms guarantee. Equipped with the knowledge that no policy regret sequences are obtainable in the game setting under reasonable play from all parties, it is natural to reason how other players would react if player $i$ deviated and what would be the cost of deviation when taking into account possible reactions.

Let us again consider the 2-player game setup through the view of player 1. The player believes their opponent might be $m$-memory bounded and decides to proceed by playing according to a no policy

regret algorithm. After many rounds of the game, player 1 has computed an empirical distribution of play $\widehat{\sigma}$ over $\mathcal{A} := \mathcal{A}_1 \times \mathcal{A}_2$. The player is familiar with the guarantees of the algorithm and knows that, if instead, they changed to playing any fixed action $a \in \mathcal{A}_1$, then the resulting empirical distribution of play $\widehat{\sigma}_a$, where player 2 has responded accordingly in a memory-bounded way, is such that $\mathbb{E}_{(a,b) \sim \widehat{\sigma}} [u_1(a,b)] \geq \mathbb{E}_{(a,b) \sim \widehat{\sigma}_a} [u_1(a,b)] - \epsilon$. This thought experiment suggests that if no policy regret play converges to an equilibrium, then the equilibrium is not only described by the deviations of player 1, but also through the change in player 2's behavior, which is encoded in the distribution $\widehat{\sigma}_a$. Thus, any equilibrium induced by no policy regret play, can be described by tuples of distributions $\{(\sigma, \sigma_a, \sigma_b) : (a,b) \in \mathcal{A}\}$, where $\sigma_a$ is the distribution corresponding to player 1's deviation to the fixed action $a \in \mathcal{A}_1$ and $\sigma_b$ captures player 2's deviation to the fixed action $b \in \mathcal{A}_2$. Clearly $\sigma_a$ and $\sigma_b$ are not arbitrary but we still need a formal way to describe how they arise.

For convenience, lets restrict the memory of player 2 to be 1. Thus, what player 1 believes is that at each round $t$ of the game, they play an action $a_t$ and player 2 plays a function $f_t : \mathcal{A}_1 \to \mathcal{A}_2$, mapping $a_{t-1}$ to $b_t = f_t(a_{t-1})$. Finally, the observed utility is $u_1(a_t, f_t(a_{t-1}))$. The empirical distribution of play, $\widehat{\sigma}$, from the perspective of player 1, is formed from the observed play $(a_t, f_t(a_{t-1}))_{t=1}^T$. Moreover, the distribution, $\widehat{\sigma}_a$, that would have occurred if player 1 chose to play action $a$ on every round is formed from the play $(a, f_t(a))_{t=1}^T$. In the view of the world of player 1, the actions taken by player 2 are actually functions rather than actions in $\mathcal{A}_2$. This suggests that the equilibrium induced by a no-policy regret play, is a distribution over the functional space defined below.

**Definition 4.1.** Let $\mathcal{F}_1 := \{f : \mathcal{A}_2^{m_1} \to \mathcal{A}_1\}$ and $\mathcal{F}_2 := \{g : \mathcal{A}_1^{m_2} \to \mathcal{A}_2\}$ denote the *functional spaces of play* of players 1 and 2, respectively. Denote the product space by $\mathcal{F} := \mathcal{F}_1 \times \mathcal{F}_2$.

Note that when $m_1 = m_2 = 0$, $\mathcal{F}$ is in a one-to-one correspondence with $\mathcal{A}$, i.e. when players believe their opponents are oblivious, we recover the action set studied in standard equilibria. For simplicity, for the remainder of the paper we assume that $m_1 = m_2 = 1$. However, all of the definitions and results that follow can be extended to the fully general setting of arbitrary $m_1$ and $m_2$; see the supplementary for details.

Let us now investigate how a distribution $\pi$ over $\mathcal{F}$ can give rise to a tuple of distributions $(\widehat{\sigma}, \widehat{\sigma}_a, \widehat{\sigma}_b)$. We begin by defining the utility of $\pi$ such that it equals the utility of a distribution $\sigma$ over $\mathcal{A}$ i.e., we want $\mathbb{E}_{(f,g) \sim \pi} [u_1(f,g)] = \mathbb{E}_{(a,b) \sim \sigma} [u_1(a,b)]$. Since utilities are not defined for functions, we need an interpretation of $\mathbb{E}_{(f,g) \sim \pi} [u_1(f,g)]$ which makes sense. We notice that $\pi$ induces a Markov chain with state space $\mathcal{A}$ in the following way.

**Definition 4.2.** Let $\pi$ be any distribution over $\mathcal{F}$. Then $\pi$ *induces a Markov process* with transition probabilities $\mathbb{P}[(a_2, b_2)|(a_1, b_1)] = \sum_{(f,g) \in \mathcal{F}_1 \times \mathcal{F}_2 : f(b_1) = a_2, g(a_1) = b_2} \pi(f,g)$. We associate with this Markov process the transition matrix $\mathrm{M} \in \mathbb{R}^{|\mathcal{A}| \times |\mathcal{A}|}$, with $\mathrm{M}_{x_1, x_2} = \mathbb{P}[x_2|x_1]$ where $x_i = (a_i, b_i)$.

Since every Markov chain with a finite state space has a stationary distribution, we think of utility of $\pi$ as the utility of a particular stationary distribution $\sigma$ of $\mathrm{M}$. How we choose $\sigma$ among all stationary distributions is going to become clear later, but for now we can think about $\sigma$ as the distribution which maximizes the utilities of both players. Next, we need to construct $\sigma_a$ and $\sigma_b$, which capture the deviation in play, when player 1 switches to action $a$ and player 2 switches to action $b$. The no-policy regret guarantee can be interpreted as $\mathbb{E}_{(f,g) \sim \pi} [u_1(f,g)] \geq \mathbb{E}_{(f,g) \sim \pi} [u_1(a, g(a))]$ i.e., if player 1 chose to switch to a fixed action (or equivalently, the constant function which maps everything to the action $a \in \mathcal{A}_1$), then their utility should not increase. Switching to a fixed action $a$, changes $\pi$ to a new distribution $\pi_a$ over $\mathcal{F}$. This turns out to be a product distribution which also induces a Markov chain.

**Definition 4.3.** Let $\pi$ be any distribution over $\mathcal{F}$. Let $\delta_a$ be the distribution over $\mathcal{F}_1$ putting all mass on the constant function mapping all actions $b \in \mathcal{A}_2$ to the fixed action $a \in \mathcal{A}_1$. Let $\pi_{\mathcal{F}_2}$ be the marginal of $\pi$ over $\mathcal{F}_2$. The *distribution resulting from player 1 switching to playing a fixed action* $a \in \mathcal{A}$, is denoted as $\pi_a = \delta_a \times \pi_{\mathcal{F}_2}$. This distribution induces a Markov chain with transition probabilities $\mathbb{P}[(a, b_2)|(a_1, b_1)] = \sum_{(f,g) : g(a_1) = b_2} \pi(f,g)$ and *the transition matrix of this Markov process is denoted by* $\mathrm{M}_a$. The distribution $\pi_b$ and matrix $\mathrm{M}_b$ are defined similarly for player 2.

Since the no policy regret algorithms we work with do not directly induce distributions over the functional space $\mathcal{F}$ but rather only distributions over the action space $\mathcal{A}$, we would like to state all of our utility inequalities in terms of distributions over $\mathcal{A}$. Thus, we would like to check if there is a stationary distribution $\sigma_a$ of $\mathrm{M}_a$ such that $\mathbb{E}_{(f,g) \sim \pi} [u_1(a, g(a))] = \mathbb{E}_{(a,b) \sim \sigma_a} [u_1(a,b)]$. This is indeed the case as verified by the following theorem.

**Theorem 4.4.** *Let $\pi$ be a distribution over the product of function spaces $\mathcal{F}_1 \times \mathcal{F}_2$. There exists a stationary distribution $\sigma_a$ of the Markov chain $\mathrm{M}_a$ for any fixed $a \in \mathcal{A}_1$ such that $\mathbb{E}_{(a,b)\sim\sigma_a}\left[u_1(a,b)\right] = \mathbb{E}_{(f,g)\sim\pi}\left[u_1(a,g(a))\right]$. Similarly, for every fixed action $b \in \mathcal{A}_2$, there exists a stationary distribution $\sigma_b$ of $\mathrm{M}_b$ such that $\mathbb{E}_{(a,b)\sim\sigma_b}\left[u_2(a,b)\right] = \mathbb{E}_{(f,g)\sim\pi}\left[u_2(f(b),b)\right]$.*

The proof of this theorem is constructive and can be found in the supplementary. With all of this notation we are ready to formally describe what no-policy regret play promises in the game setting in terms of an equilibrium.

**Definition 4.5.** A distribution $\pi$ over $\mathcal{F}_1 \times \mathcal{F}_2$ is a *policy equilibrium* if for all fixed actions $a \in \mathcal{A}_1$ and $b \in \mathcal{A}_2$, which generate Markov chains $\mathrm{M}_a$ and $\mathrm{M}_b$ respectively, with stationary distributions $\sigma_a$ and $\sigma_b$ from Theorem 4.4, there exists a stationary distribution $\sigma$ of the Markov chain $\mathrm{M}$ induced by $\pi$ such that:

$$\mathbb{E}_{(a,b)\sim\sigma}\left[u_1(a,b)\right] \geq \mathbb{E}_{(a,b)\sim\sigma_a}\left[u_1(a,b)\right]$$
$$\mathbb{E}_{(a,b)\sim\sigma}\left[u_2(a,b)\right] \geq \mathbb{E}_{(a,b)\sim\sigma_b}\left[u_2(a,b)\right]. \tag{3}$$

In other words, $\pi$ is a policy equilibrium if there exists a stationary distribution $\sigma$ of the Markov chain corresponding to $\pi$, such that, when actions are drawn according to $\sigma$, no player has incentive to change their action. For a simple example of a policy equilibrium see Section E in the supplementary.

## 4.1 Convergence to the set of policy equilibria

We have tried to formally capture the notion of equilibria in which player 1's deviation would lead to a reaction from player 2 and vice versa in Definition 4.5. This definition is inspired by the counterfactual guarantees of no policy regret play and we would like to check that if players' strategies yield sublinear policy regret then the play converges to a policy equilibrium. Since the definition of sublinear policy regret does not include a distribution over functional spaces but only works with empirical distributions of play, we would like to present our result in terms of distributions over the action space $\mathcal{A}$. Thus we begin by defining the set of all product distributions $\sigma \times \sigma_a \times \sigma_b$, induced by policy equilibria $\pi$ as described in the previous subsection. Here $\sigma_a$ and $\sigma_b$ represent the deviation in strategy if player 1 changed to playing the fixed action $a \in \mathcal{A}_1$ and player 2 changed to playing the fixed action $b \in \mathcal{A}_2$ respectively as constructed in Theorem 4.4.

**Definition 4.6.** For a policy equilibrium $\pi$, let $S_\pi$ be the set of all stationary distributions which satisfy the equilibrium inequalities (3), $S_\pi := \{\sigma \times \sigma_a \times \sigma_b : (a,b) \in \mathcal{A}\}$. Define $S = \bigcup_{\pi \in \Pi} S_\pi$, where $\Pi$ is the set of all policy equilibria.

Our main result states that the sequence of empirical product distributions formed after $T$ rounds of the game $\widehat{\sigma} \times \widehat{\sigma}_a \times \widehat{\sigma}_b$ is going to converge to $S$. Here $\widehat{\sigma}_a$ and $\widehat{\sigma}_b$ denote the distributions of deviation in play, when player 1 switches to the fixed action $a \in \mathcal{A}_1$ and player 2 switches to the fixed action $b \in \mathcal{A}_2$ respectively. We now define these distributions formally.

**Definition 4.7.** Suppose player 1 is playing an algorithm with output at time $t$ given by $f_t : \mathcal{A}_2^t \to \Delta\mathcal{A}_1$ i.e. $p_t^1 = f_t(b_{0:t-1})$. Similarly, suppose player 2 is playing an algorithm with output at time $t$ given by $p_t^2 = g_t(a_{0:t-1})$. The empirical distribution at time $T$ is $\widehat{\sigma} := \frac{1}{T}\sum_{t=1}^{T} p_t$, where $p_t = p_t^1 \times p_t^2$ is the product distribution over $\mathcal{A}$ at time $t$. Further let $(p_a^2)_t = g_t(a_{0:t-m}, a, \ldots, a)$ denote the distribution at time $t$, provided that player 1 switched their strategy to the constant action $a \in \mathcal{A}_1$. Let $\delta_a$ denote the distribution over $\mathcal{A}_1$ which puts all the probability mass on action $a$. Let $(p_a)_t = \delta_a \times (p_a^2)_t$ be the product distribution over $\mathcal{A}$, corresponding to the change of play at time $t$. Denote by $\widehat{\sigma}_a = \frac{1}{T}\sum_{t=1}^{T}(p_a)_t$ the empirical distribution corresponding to the change of play. The distribution $\widehat{\sigma}_b$ is defined similarly.

Suppose that $f_t$ and $g_t$ are no-policy regret algorithms, then our main result states that the sequence $(\widehat{\sigma} \times \widehat{\sigma}_a \times \widehat{\sigma}_b)_T$ converges to the set $S$.

**Theorem 4.8.** *If the algorithms played by player 1 in the form of $f_t$ and player 2 in the form of $g_t$ give sub-linear policy regret sequences, then the sequence of product distributions $(\widehat{\sigma} \times \widehat{\sigma}_a \times \widehat{\sigma}_b)_{T=1}^{\infty}$ converges weakly to the set $S$.*

In particular if both players are playing MWU or Exp3, we know that they will have sublinear policy regret. Not surprisingly, we can show something slightly stronger as well. Let $\tilde{\sigma}$, $\tilde{\sigma}_a$ and $\tilde{\sigma}_b$ denote the empirical distributions of observed play corresponding to $\widehat{\sigma}$, $\widehat{\sigma}_a$ and $\widehat{\sigma}_b$, i.e. $\tilde{\sigma} = \frac{1}{T}\delta_t$, where $\delta_t$

denotes the Dirac distribution, putting all weight on the played actions at time $t$. Then these empirical distributions also converge to $S$ almost surely.

## 4.2 Sketch of proof of the main result

The proof of Theorem 4.8 has three main steps. The first step defines the natural empirical Markov chains $\widehat{M}$, $\widehat{M}_a$ and $\widehat{M}_b$ from the empirical play $(p_t)_{t=1}^\infty$ (see Definition B.2) and shows that the empirical distributions $\widehat{\sigma}$, $\widehat{\sigma}_a$ and $\widehat{\sigma}_b$ are stationary distributions of the respective Markov chains. The latter is done in Lemma B.3. The next step is to show that the empirical Markov chains converge to Markov chains $M$, $M_a$ and $M_b$ induced by some distribution $\pi$ over $\mathcal{F}$. In particular, we construct an empirical distribution $\widehat{\pi}$ and distributions $\widehat{\pi}_a$ and $\widehat{\pi}_b$ corresponding to player's deviations (see Definition B.5), and show that these induce the Markov chains $\widehat{M}$, $\widehat{M}_a$ and $\widehat{M}_b$ respectively (Lemma B.7). The distribution $\pi$ we want is now the limit of the sequence $(\widehat{\pi})_T$. The final step is to show that $\pi$ is a policy equilibrium. The proof goes by contradiction. Assume $\pi$ is not a policy equilibrium, this implies that no stationary distribution of $M$ and corresponding stationary distributions of $M_a$ and $M_b$ can satisfy inequalities (3). Since the empirical distributions $\widehat{\sigma}$, $\widehat{\sigma}_a$ and $\widehat{\sigma}_b$ of the play satisfies inequalities (3) up to an $o(1)$ additive factor, we can show, in Theorem B.8, that in the limit, the policy equilibrium inequalities are exactly satisfied. Combined with the convergence of $\widehat{M}$, $\widehat{M}_a$ and $\widehat{M}_b$ to $M$, $M_a$ and $M_b$, respectively, this implies that stationary distributions of $M$, $M_a$ and $M_b$, satisfying (3), giving a contradiction.

We would like to emphasize that the convergence guarantee of Theorem 4.8 does not rely on there being a unique stationary distribution of the empirical Markov chains $\widehat{M}$, $\widehat{M}_a$ and $\widehat{M}_b$ or their respective limits $M, M_a, M_b$. Indeed, Theorem 4.8 shows that any limit point of $\{(\widehat{\sigma}, \widehat{\sigma}_a, \widehat{\sigma}_b)_T\}_{T=1}^\infty$ satisfies the conditions of Definition 4.5. The proof does not require that any of the respective Markov chains have a unique stationary distribution, but rather requires only that $\widehat{\sigma}$ has sublinear policy regret. We would also like to remark that $\{(\widehat{\sigma}, \widehat{\sigma}_a, \widehat{\sigma}_b)_T\}_{T=1}^\infty$ need not have a unique limit and our convergence result only guarantees that the sequence is going to the set $S$. This is standard when showing that any type of no regret play converges to an equilibrium, see for example Stoltz and Lugosi (2007).

## 4.3 Relation of policy equlibria to CCEs

So far we have defined a new class of equilibria and shown that they correspond to no policy regret play. Furthermore, we know that if both players in a 2-player game play stable no external regret algorithms, then their play also has sublinear policy regret. It is natural to ask if every CCE is also a policy equilibrium: if $\sigma$ is a CCE, is there a corresponding policy equilibrium $\pi$ which induces a Markov chain $M$ for which $\sigma$ is a stationary distribution satisfying (3)? We show that the answer to this question is positive:

**Theorem 4.9.** *For any CCE $\sigma$ of a 2-player game $\mathcal{G}$, there exists a policy-equilibrium $\pi$ which induces a Markov chain $M$ with stationary distribution $\sigma$.*

To prove this, we show that for any CCE we can construct stable no-external regret algorithm which converge to it, and so since stable no-external regret algorithms always converge to policy equilibria (Theorem 3.4), this implies the CCE is also a policy equilibrium.

However, we show the converse is not true: policy equilibria can give rise to behavior which is not a CCE. Our proof appeals to a utility sequence which is similar in spirit to the one in Theorem 3.2, but is adapted to the game setting.

**Theorem 4.10.** *There exists a 2-player game $\mathcal{G}$ and product distributions $\sigma \times \sigma_a \times \sigma_b \in S$ (where $S$ is defined in Definition 4.6 as the possible distributions of play from policy equilibria), such that $\sigma$ is not a CCE of $\mathcal{G}$.*

In Section E of the supplementary we give a simple example of a policy equilibrium which is not a CCE.

# 5 Discussion

In this work we gave a new twist on policy regret by examining it in the game setting, where we introduced the notion of policy equilibrium and showed that it captures the behavior of no policy

regret players. While our characterization is precise, we view this as only the first step towards truly understanding policy regret and its variants in the game setting. Many interesting open questions remain. Even with our current definitions, since we now have a broader class of equilibria to consider it is natural to go back to the extensive literature in algorithmic game theory on the price of anarchy and price of stability and reconsider it in the context of policy equilibria. For example Roughgarden (2015) showed that in "smooth games" the worst CCE is no worse than the worst Nash. Since policy equilibria contain all CCEs (Theorem 4.9), is the same true for policy equilibria?

Even more interesting questions remain if we change our definitions to be more general. For example, what happens with more than 2 players? With three or more players, definitions of "reaction" by necessity become more complicated. Or what happens when $m$ is not a constant? No policy regret algorithms exist for superconstant $m$, but our notion of equilibrium requires $m$ to be constant in order for the Markov chains to make sense. Finally, what if we compare against deviations that are more complicated than a single action, in the spirit of swap regret or $\Phi$-regret?

From an online learning perspective, note that our notion of on average stable and the definition of $m$-memory boundedness are different notions of stability. Is there one unified definition of "stable" which would allow us to give no policy regret algorithms against stable adversaries even outside of the game setting?

### Acknowledgments

This work was supported in part by NSF BIGDATA grant IIS-1546482, NSF BIGDATA grant IIS-1838139, NSF CCF-1535987, NSF IIS-1618662, NSF CCF-1464239, and NSF AITF CCF-1535887.

## Footnotes

[1]Such games can be equivalently described in terms of minimizing losses rather than maximizing utilities. All our results can be equivalently expressed in terms of losses instead of utilities.

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
