[Supplementary Material · supplementary.pdf]

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

# Supplementary Material to "Policy Regret in Repeated Games"

## A   Proofs of results from Section 3

**Theorem A.1.** *Let the space of actions be $\{0, 1\}$ and define the $m$-memory bounded utility functions as follows:*

$$f_t(a_{t-m+1}, .., a_t) = \begin{cases} 1 & a_{t-m+i} = a_{t-m+i+1} = 1 \text{ for } i \in \{1, .., m-2\} \wedge a_{t-1} \neq a_t \\ \frac{1}{2} & a_{t-m+i} = a_{t-m+i+1} = 1 \text{ for } i \in \{1, .., m-1\} \\ 0 & \text{otherwise} \end{cases} .$$

*Assuming $m \geq 3$ is a fixed constant (independent of $T$) any sequence with with sublinear policy regret will have linear external regret and every sequence with sublinear external regret will have linear policy regret.*

*Proof of Theorem 3.2.* Let $(a_t)_{t=1}^T$ be a sequence with sublinear policy regret. Then this sequence has utility at least $\frac{T}{2} - o(T)$ and so there are at most $o(T)$ terms in the sequence which are not equal to $1$. Let the subsequence consisting of all $a_t = 0$ be indexed by $\mathcal{I}$. Define $\tilde{\mathcal{I}} = \{t, t + 1, \cdots, t + m - 1 : t \in \mathcal{I}\}$ and consider the subsequence of functions $(f_t)_{t \notin \tilde{\mathcal{I}}}$. This is precisely the sequence of functions which have utility $\frac{1}{2}$ with respect to the sequence of play $(a_t)_{t=1}^T$. Notice that the length of this sequence is at least $T - mo(T) = T - o(T)$. The utility of this sequence is $\sum_{t \notin \tilde{\mathcal{I}}} -f_t(a_{t-m+1}, .., a_t) \sum_{t \notin \tilde{\mathcal{I}}} f_t(1, .., 1) = \frac{T - o(T)}{2}$, however, this subsequence has linear external regret, since $\sum_{t \notin \tilde{\mathcal{I}}} f_t(a_{t-m+1}, .., a_{t-1}, 0) = \sum_{t \notin \tilde{\mathcal{I}}} f_t(1, .., 1, 0) = T - o(T)$. Thus the external regret of $(a_t)_{t=1}^T$ is

$$\begin{aligned}
\sum_{t=1}^T [f_t(a_{t-m+1}, .., 0) - f_t(a_{t-m+1}, .., a_t)] &= \sum_{t \notin \tilde{\mathcal{I}}} [f_t(a_{t-m+1}, .., 0) - f_t(a_{t-m+1}, .., a_t)] \\
&+ \sum_{t \in \tilde{\mathcal{I}}} [f_t(a_{t-m+1}, .., 0) - f_t(a_{t-m+1}, .., a_t)] \\
&\geq \frac{T - o(T)}{2} + \sum_{t \in \tilde{\mathcal{I}}} [f_t(a_{t-m+1}, .., 0) - f_t(a_{t-m+1}, .., a_t)] \\
&\geq \frac{T}{2} - o(T),
\end{aligned}$$

where the last inequality follows from the fact that the cardinality of $\tilde{\mathcal{I}}$ is at most $o(T)$ and thus $\sum_{t \in \tilde{\mathcal{I}}} [f_t(a_{t-m+1}, .., 0) - f_t(a_{t-m+1}, .., a_t)] \geq -o(T)$.

Assume that $(a_t)_{t=1}^T$ has sublinear external regret. From the above argument, it follows that the utility of the sequence is at most $o(T)$ (otherwise if the sequence has utility $\omega(T)$, we can repeat the previous argument and get a contradiction with the fact the sequence is no-external regret). This implies that the the policy regret of the sequence is $\sum_{t=1}^T [f_t(1, 1, \cdots, 1) - f_t(a_{t-m+1}, .., a_t)] = \frac{T}{2} - o(T)$.  □

**Theorem A.2.** *Let $(a_t)_{t=1}^T$ and $(b_t)_{t=1}^T$ be the action sequences of player $1$ and $2$ respectively and suppose that they are coming from no-regret algorithms with regrets $R_1(T)$ and $R_2(T)$ respectively. Assume that the algorithms are on average $(m, S(T))$ stable with respect to the $\ell_2$ norm. Then*

$$\mathbb{E}\left[ u_2(f_t(b_{0:t-m+1}, b, \cdots, b), b) - \sum_{t=1}^T u_2(f_t(b_{0:t-1}), b_t) \right] \leq \|\mathrm{P}_2\| S(T) + R_2(T),$$

*for any fixed action $b \in \mathcal{A}_2$, where $\mathrm{P}_2$ is the utility matrix of player $2$. A similar inequality holds for any fixed action $a \in \mathcal{A}_1$ and the utility of player $1$.*

*Proof of Theorem 3.4.*

$$\mathbb{E}\left[u_2(f_t(b_0,\cdots,b_{t-m+1},b,\cdots,b),b) - \sum_{t=1}^{T} u_2(f_t(b_0,\cdots,b_{t-2},b_{t-1}),b_t)\right]$$

$$= \mathbb{E}\left[u_2(f_t(b_0,\cdots,b_{t-m+1},b,\cdots,b),b) - \sum_{t=1}^{T} u_2(f_t(b_0,\cdots,b_{t-2},b_{t-1}),b)\right]$$

$$+ \mathbb{E}\left[u_2(f_t(b_0,\cdots,b_{t-1}),b) - \sum_{t=1}^{T} u_2(f_t(b_0,\cdots,b_{t-2},b_{t-1}),b_t)\right]$$

$$\leq \sum_{t=1}^{T} \|b^\top \mathrm{P}_2\|_2 \|f_t(b_0,\cdots,b_{t-m+1},b,\cdots,b) - f_t(b_0,\cdots,b_{t-2},b_{t-1})\|_2 + R_2(T)$$

$$\leq \|b\|_1 \|\mathrm{P}_2\| S(T) + R_2(T) = \|\mathrm{P}_2\| S(T) + R_2(T)$$

where the first inequality holds by Cauchy-Schwartz, the second inequality holds by using the $m$-stability of the algorithm, together with the inequality between $l_1$ and $l_2$ norms. $\qquad\square$

The next theorems show that MWU and Exp3 are stable algorithms.

**Theorem A.3.** *MWU is an on average $(m, m\sqrt{T})$ stable algorithm with respect to $\ell_1$, for any $m < o(\sqrt{T})$.*

*Proof of Theorem A.3.* We think of MWU as Exponentiated Gradient (EG) where the loss vector has $i$-th entry equal to the negative utility if the player decided to play action $i$. Let the observed loss vector at time $j$ be $\widehat{l}_j$ and the output distribution be $p_j$, then the update of EG can be written as $p_{j+1} = \arg\min_{p \in \mathcal{C}} \langle \widehat{l}_j, p \rangle + \frac{1}{\eta} D(p, p_j)$, where $C$ is the simplex of the set of possible actions and $D$ is the KL-divergence. Using Lemma 3 in Saha et al. (2012), with $f(p) = \langle \widehat{l}_j, p \rangle + \frac{1}{\eta} D(p, p_j)$ and the fact that the KL-divergence is 1-strongly convex over the simplex $\mathcal{C}$ with respect to the $\ell_1$ norm, we have:

$$\frac{1}{2\eta}\|p_j - p_{j+1}\|_1^2 \leq f(p_j) - f(p_{j+1}) = \langle p_j - p_{j+1}, \widehat{l}_j \rangle - \frac{1}{\eta} D(p_{j+1}, p_j)$$

$$\leq \|p_j - p_{j+1}\|_1 \|\widehat{l}_j\|_\infty \leq \|p_j - p_{j+1}\|_1,$$

where the second inequality follows from Hölder's inequality and the fact that $D(p_{j+1}, p_j) \geq 0$. Thus with step size $\eta \sim \sqrt{\frac{1}{T}}$, we have $\|p_j - p_{j+1}\|_1 \leq \frac{1}{2\sqrt{T}}$. Using triangle inequality, we can get $\|p_{j-1} - p_{j+1}\|_1 \leq \|p_{j-1} - p_j\|_1 + \|p_j - p_{j+1}\|_1 \leq \frac{2}{2\sqrt{T}}$ and induction shows that $\|p_{j-m+1} - p_{j+1}\|_1 \leq \frac{m}{2\sqrt{T}}$. Suppose for the last $m$ iterations, a fixed loss function $l_a$ was played instead and the resulting output of the algorithm becomes $\tilde{p}_{j+1}$. Then using the same argument as above we have $\|p_{j-m+1} - \tilde{p}_{j+1}\|_1 \leq \frac{m}{2\sqrt{T}}$ and thus $\|p_{j+1} - \tilde{p}_{j+1}\|_1 \leq \frac{m}{\sqrt{T}}$. Summing over all $T$ rounds concludes the proof. $\qquad\square$

**Theorem A.4.** *Exp3 is an on average $(m, m\sqrt{T})$ stable algorithm with respect to $\ell_1$, for any $m < o(\sqrt{T})$*

*Proof of Theorem A.4.* The update at time $t$, conditioning on the the draw being $i$, is given by $p_{t+1}^i = \frac{w_t^i \exp\left(\frac{\gamma}{kp_t^i} u_t^i\right)}{w_t^i \exp\left(\frac{\gamma}{kp_t^i} u_t^i\right) + \sum_{j \neq i} w_t^j}$ and for $j \neq i$, $p_{t+1}^j = \frac{w_t^j}{w_t^i \exp\left(\frac{\gamma}{kp_t^i} u_t^i\right) + \sum_{j \neq i} w_t^j}$, where $u_t$ is the utility vector at time $t$, $w_t$ is the weight vector at time $t$ i.e. $w_{t+1}^i = w_t^i \exp\left(\frac{\gamma}{kp_t^i} u_t^i\right)$, $w_{t+1}^j = w_t^j$, and $k$ is the number of actions. We have the following bound:

$$|p_{t+1}^i - p_t^i| = \left|\frac{w_t^i \exp\left(\frac{\gamma}{kp_t^i} u_t^i\right)}{w_t^i \exp\left(\frac{\gamma}{kp_t^i} u_t^i\right) + \sum_{j \neq i} w_t^j} - \frac{w_t^i}{\sum_j w_t^j}\right| \leq \left|\frac{w_t^i(\exp\left(\frac{\gamma}{kp_t^i} u_t^i\right) - 1)}{\sum_j w_t^j}\right|$$

$$= p_t^i(\exp\left(\frac{\gamma}{kp_t^i} u_t^i\right) - 1) \leq p_t^i 2\frac{\gamma}{kp_t^i} u_t^i \leq 2\frac{\gamma}{k},$$

where the first inequality uses the fact that $p_{t+1}^i \geq p_t^i$ and the second inequality uses the choice of $\gamma$ together with $\exp(x) \leq 2x + 1$ for $x \in [0, 1]$. Similarly for $j \neq i$, we have:

$$|p_{t+1}^j - p_t^j| = \left| \frac{w_t^j}{w_t^i \exp\left(\frac{\gamma}{k p_t^i} u_t^i\right) + \sum_{j \neq i} w_t^j} - \frac{w_t^j}{\sum_j w_t^j} \right| \leq \left| \frac{w_t^j}{\sum_j w_t^j} \left(1 - \frac{1}{\exp\left(\frac{\gamma u_t^i}{k p_t^i}\right)}\right) \right|$$

$$= p_t^j \left(1 - \exp\left(-\frac{\gamma u_t^i}{k p_t^i}\right)\right) \leq p_t^j \frac{\gamma u_t^i}{k p_t^i} \leq \frac{p_t^j}{k p_t^i} \gamma,$$

where we have used $p_{t+1}^j \leq p_t^j$ and $\exp(-x) \geq 1 - x$ for all $x$. We can now proceed to bound $\mathbb{E}_{i_t}\left[\|p_{t+1} - p_t\|_1 | i_{1:t-1}\right]$, where $i_t$ is the random variable denoting the draw at time $t$:

$$\mathbb{E}_{i_t}\left[\|p_{t+1} - p_t\|_1 | i_{1:t-1}\right] = \sum_i p_t^i \|p_{t+1} - p_t\|_1 = \sum_i p_t^i \left(|p_{t+1}^i - p_t^i| + \sum_{j \neq i} |p_{t+1}^j - p_t^j|\right)$$

$$\leq 2\gamma + \sum_{i,j} p_t^i \frac{p_t^j}{k p_t^i} \gamma = 3\gamma.$$

Setting $\gamma \sim \frac{1}{\sqrt{T}}$ finishes the proof. $\qquad\square$

Combining the above results together, we can show that both players will also have no-policy regret for any $m < o(\sqrt{T})$.

**Corollary A.5.** *Let $(a_t)_{t=1}^T$ and $(b_t)_{t=1}^T$, be the action sequences of players $1$ and $2$ respectively and suppose that the sequences are coming from MWU. Then for any fixed $m$, it holds:*

$$\mathbb{E}\left[u_2(f_t(b_{0:t-m+1}, b, \cdots, b), b) - \sum_{t=1}^T u_2(f_t(b_{0:t-1}), b_t)\right] \leq O(\|P_2\| m\sqrt{T}),$$

*for any fixed action $b \in \mathcal{A}_2$, where $f_t$ are the functions corresponding to the MWU algorithm used by player $1$.*

*Proof.* From Theorem A.3 it follows that MWU is on average $(m, m\sqrt{T})$ stable and has regret at most $O(\sqrt{T})$. $\qquad\square$

# B   Proofs of results from Section 4

**Theorem B.1.** *Let $\pi$ be a distribution over the product of function spaces $\mathcal{F}_1 \times \mathcal{F}_2$. There exists a stationary distribution $\tilde{\sigma}$ of the Markov chain $M_a$ for any fixed $a \in \mathcal{A}_1$ and $(f, g) \sim \pi$ such that $\mathbb{E}_{(a,b) \sim \tilde{\sigma}}[u_1(a, b)] = \mathbb{E}_{(f,g) \sim \pi}[u_1(a, g(a))]$*

*Proof of Theorem 4.4.* Note that, by definition $(M_a)_{(\tilde{a},\tilde{b}),(\widehat{a},\widehat{b})} = 0$ if $\widehat{a} \neq a$ and $M_{(\tilde{a},\tilde{b}),(\widehat{a},\widehat{b})} = \sum_{f,g:g(\tilde{a})=\widehat{b}} \pi(f, g)$ if $\widehat{a} = a$. Consider the distribution $\tilde{\sigma}$ over $\mathcal{A}$, where $\tilde{\sigma}_{(\tilde{a},\tilde{b})} = 0$ if $\tilde{a} \neq a$ and $\tilde{\sigma}_{(\tilde{a},\tilde{b})} = \sum_{f,g:g(a)=\tilde{b}} \pi(f, g)$ if $\tilde{a} = a$. We now show that $\tilde{\sigma}$ is a stationary distribution of $M_a$:

$$\left(\tilde{\sigma}^\top M_a\right)_{(a,\tilde{b})} = \sum_{(\widehat{a},\widehat{b})} \tilde{\sigma}_{(\widehat{a},\widehat{b})} (M_a)_{(\widehat{a},\widehat{b}),(a,\tilde{b})} = \sum_{\widehat{b}} \tilde{\sigma}_{(a,\widehat{b})} (M_a)_{(a,\widehat{b}),(a,\tilde{b})}$$

$$= \sum_{\widehat{b}} \left(\sum_{f,g:g(a)=\widehat{b}} \pi(f, g)\right)\left(\sum_{f,g:g(a)=\tilde{b}} \pi(f, g)\right)$$

$$= \left(\sum_{f,g:g(a)=\tilde{b}} \pi(f, g)\right)\left(\sum_{\widehat{b}} \sum_{f,g:g(a)=\widehat{b}} \pi(f, g)\right)$$

$$= \sum_{f,g:g(a)=\tilde{b}} \pi(f, g) = \tilde{\sigma}_{(a,\tilde{b})}.$$

Finally, notice that:

$$\mathbb{E}_{(f,g)\sim\pi}\left[u_1(a,g(a))\right] = \sum_{b\in\mathcal{A}_2} u_1(a,b)\mathbb{P}\left[g(a)=b\right] = \sum_{b\in\mathcal{A}_2} u_1(a,b) \sum_{f,g:g(a)=b}\pi(f,g).$$

□

## B.1 Sketch of proof of the main result

The proof of Theorem 4.8 has three main steps. The first step defines the natural empirical Markov chains $\widehat{\mathrm{M}}$, $\widehat{\mathrm{M}}_a$ and $\widehat{\mathrm{M}}_b$ from the empirical play $(p_t)_{t=1}^t$ and shows that the empirical distributions $\widehat{\sigma}$, $\widehat{\sigma}_a$ and $\widehat{\sigma}_b$ are stationary distributions of the respective Markov chains. The next step is to show that the empirical Markov chains converge to Markov chains $\mathrm{M}$, $\mathrm{M}_a$ and $\mathrm{M}_b$ induced by some distribution $\pi$ over $\mathcal{F}$. The final step is to show that $\pi$ is a policy equilibrium.

We begin with the definition of the empirical Markov chains.

**Definition B.2.** Let the empirical Markov chain be $\widehat{\mathrm{M}}$, with $\widehat{\mathrm{M}}_{i,j} = \frac{\frac{1}{T}\sum_{t=1}^T p_t(x_i)p_t(x_j)}{\frac{1}{T}\sum_{t=1}^T p_t(x_i)}$ if $\frac{1}{T}\sum_{t=1}^T p_t(x_i) \neq 0$ and $0$ otherwise, where $p_t$ is defined in 4.7. For any fixed $a \in \mathcal{A}_1$, let the empirical Markov chain corresponding to the deviation in play of player 1 be $\widehat{\mathrm{M}}_a$, with $(\widehat{\mathrm{M}}_a)_{i,j} = \frac{\frac{1}{T}\sum_{t=1}^T (p_a)_t(x_i)(p_a)_t(x_j)}{\frac{1}{T}\sum_{t=1}^T (p_a)_t(x_i)}$, if $\frac{1}{T}\sum_{t=1}^T (p_a)_t(x_i) \neq 0$ and $0$ otherwise, where $(p_a)_t$ is defined in 4.7. The Markov chain $\widehat{\mathrm{M}}_b$ is defined similarly for any $b \in \mathcal{A}_2$.

The intuition behind constructing these Markov chains is as follows – if we were only provided with the observed empirical play $(x_t)_{t=1}^T = (a_t, b_t)_{t=1}^T$ and someone told us that the $x_t$'s were coming from a Markov chain, we could try to build an estimator of the Markov chain by approximating each of the transition probabilities. In particular the estimator of transition from state $i$ to state $j$ is given by $\tilde{\mathrm{M}}_{i,j} = \frac{\frac{1}{T}\sum_{t=1}^T \delta_{t-1}(x_i)\delta_t(x_j)}{\frac{1}{T}\sum_{t=1}^T \delta_t(x_i)}$, where $\delta_t(x_i) = 1$ if $x_i$ occurred at time $t$ and $0$ otherwise. When the players are playing according to a no-regret algorithm i.e. at time $t$, $x_t$ is sampled from $p_t$, it is possible to show that $\tilde{\mathrm{M}}_{i,j}$ concentrates to $\widehat{\mathrm{M}}_{i,j}$ (see section B.4). Not only does $\widehat{\mathrm{M}}$ arise naturally, but it turns out that the empirical distribution $\widehat{\sigma}$ defined in 4.7 is also a stationary distribution of $\widehat{\mathrm{M}}$.

**Lemma B.3.** *The distribution of play $\widehat{\sigma} = \frac{1}{T}\sum_{t=1}^T p_t$ is a stationary distribution of $\widehat{\mathrm{M}}$. Similarly the distributions $\tilde{\sigma}, \widehat{\sigma}_a, \tilde{\sigma}_a, \widehat{\sigma}_b$ and $\tilde{\sigma}_b$ are stationary distributions of the Markov chains $\tilde{\mathrm{M}}, \widehat{\mathrm{M}}_a, \tilde{\mathrm{M}}_a, \widehat{\mathrm{M}}_b$ and $\tilde{\mathrm{M}}_b$ respectively.*

*Proof.* We show the result for $\widehat{\sigma}$ and $\widehat{\mathrm{M}}$. The rest of the results can then be derived in the same way.

$$(\widehat{\sigma}^{\top}\widehat{\mathrm{M}})_j = \sum_{i=1}^{|\mathcal{A}|}\left(\frac{1}{T}\sum_{t=1}^T p_t(x_i)\right)\frac{\frac{1}{T}\sum_{t=1}^T p_t(x_i)p_t(x_j)}{\frac{1}{T}\sum_{t=1}^T p_t(x_i)} = \frac{1}{T}\sum_{t=1}^T p_t(x_j)\sum_{i=1}^{|\mathcal{A}|} p_t(x_i) = \frac{1}{T}\sum_{t=1}^T p_t(x_j),$$

where the first equality holds because the $i$-th entry of the vector $\widehat{\sigma}$ is exactly $\frac{1}{T}\sum_{t=1}^T p_t(x_i)$ and the $(i,j)$-th entry of $\widehat{\mathrm{M}}$ by definition is $\frac{\frac{1}{T}\sum_{t=1}^T p_t(x_i)p_t(x_j)}{\frac{1}{T}\sum_{t=1}^T p_t(x_i)}$, and the last equality holds because $\mathrm{p}_t$ is a distribution over actions so $\sum_{i=1}^{|\mathcal{A}|} p_t(x_i) = 1$. □

Suppose that both players are playing MWU for $T$ rounds. Then Lemma B.3 together with Theorems 3.4 and the stability of MWU imply that $\mathbb{E}_{(a,b)\sim\widehat{\sigma}}\left[u_1(a,b)\right] \geq \mathbb{E}_{(a,b)\sim\widehat{\sigma}_a}\left[u_1(a,b)\right] - O(m/\sqrt{T})$. A similar inequality holds for player 2 and $\widehat{\sigma}_b$. As $T \to \infty$, the inequality above becomes similar to (3). This will play a crucial role in the proof of our convergence result, which shows that $\widehat{\sigma}, \widehat{\sigma}_a$ and $\widehat{\sigma}_b$ converge to the set of policy equilibria. We would also like to guarantee that the empirical distributions of observed play $\tilde{\sigma}, \tilde{\sigma}_a$ and $\tilde{\sigma}_b$ also converge to this set. To show this second result, we are going to proof that $\tilde{\sigma}$ approaches $\widehat{\sigma}$ almost surely as $T$ goes to infinity.

**Lemma B.4.** *Let $\widehat{\sigma} = \frac{1}{T}\sum_{t=1}^T p_t$ be the empirical distribution after $T$ rounds of the game and let $\tilde{\sigma} = \frac{1}{T}\sum_{t=1}^T \delta_t$ be the empirical distribution of observed play. Then $\limsup_{T\to\infty}\|\tilde{\sigma} - \widehat{\sigma}\|_1 = 0$ almost surely. Similarly, for the distributions corresponding to deviation in play we have $\limsup_{T\to\infty}\|\tilde{\sigma}_a - \widehat{\sigma}_a\|_1 = 0$ and $\limsup_{T\to\infty}\|\tilde{\sigma}_b - \widehat{\sigma}_b\|_1 = 0$ almost surely.*

*Proof.* The proof follows the same reasoning as the proof of lemma B.19. □

Our next step is to show that the empirical Markov chains $\widehat{\mathrm{M}}$ converge to a Markov chain $\mathrm{M}$ induced by some distribution $\pi$ over the functional space $\mathcal{F}$. We do so by constructing a sequence of empirical distributions $\widehat{\pi}$ over $\mathcal{F}$, based on the players' strategies, which induce $\widehat{\mathrm{M}}$. We can then consider every convergent subsequence of $(\widehat{\pi})_{T_\ell=1}^\infty$ with limit point $\pi$ and argue that the corresponding sequence $(\widehat{\mathrm{M}})_{T_\ell=1}^\infty$ of Markov chains converges to the Markov chain induced by $\pi$.

**Definition B.5.** Let $\widehat{\pi}$ be the distribution over $\mathcal{F}$, such that the probability to sample any fixed $f : \mathcal{A}_2 \to \mathcal{A}_1$ and $g : \mathcal{A}_1 \to \mathcal{A}_2$ is $\widehat{\pi}(f,g) = \prod_{i \in |\mathcal{A}|} \frac{\sum_t p_t(x_i)p_t(y_i)}{\sum_{t=1} p_t(x_i)}$ , where $x_i = (a_i, b_i)$ and $y_i = (f(b_i), g(a_i))$. Similarly, let $\widehat{\pi}_a$ and $\widehat{\pi}_b$ be the distributions over $\mathcal{F}$ constructed as above but by using the empirical distribution of deviated play induced by player 1 deviating to action $a \in \mathcal{A}_1$ and player 2 deviating to action $b \in \mathcal{A}_2$.

The next lemma checks that $\widehat{\pi}$ is really a probability distribution.

**Lemma B.6.** *The functionals $\widehat{\pi}, \widehat{\pi}_a$ and $\widehat{\pi}_b$ are all probability distributions.*

*Proof.* Consider the space of all transition events for a fixed $(a,b)$ pair i.e. $\mathcal{S}_{(a,b)} = \{((a',b') \times (a,b)) : (a',b') \in \mathcal{A}\}$. There is an inherent probability measure on this set, given by $\mathbb{P}[(a',b') \times (a,b)] = \frac{\sum_t p_t(a,b)p_t(a',b')}{\sum_t p_t(a,b)}$. It is easy to see that this is a probability measure, since the measure of the whole set is exactly

$$\sum_{(a',b') \in \mathcal{A}} \frac{\sum_t p_t(a,b)p_t(a',b')}{\sum_t p_t(a,b)} = \frac{\sum_t p_t(a,b) \sum_{(a',b') \in \mathcal{A}} p_t(a',b')}{\sum_t p_t(a,b)} = 1.$$

The set of all $\mathcal{F}$ can exactly be thought of as $\times_{(a,b) \in \mathcal{A}} \mathcal{S}_{(a,b)}$ and the function $\widehat{\pi}$ defined in B.5 is precisely the product measure on that set. Similar arguments show that $\widehat{\pi}_a$ and $\widehat{\pi}_b$ are probability distributions. □

The proof of the above lemma reveals something interesting about the construction of $\widehat{\pi}$. Fix the actions $(a,b) \in \mathcal{A}$. Then the probability to sample a function pair $(f,g)$ which map $(a,b)$ to $(a',b')$ i.e. $a' = f(a)$ and $b' = f(b)$ is precisely equal to the entry $\widehat{\mathrm{M}}_{(a,b),(a',b')}$ of the empirical Markov chain. Since every function pair $(f,g) \in \mathcal{F}$ is determined by the way $\mathcal{A}$ is mapped, and we have already have a probability distribution for a fixed mapping $(a,b)$ to $(a',b')$, we can just extend this to $\widehat{\pi}$ by taking the product distribution over all pairs $(a,b) \in \mathcal{A}$. This construction gives us exactly that $\widehat{\mathrm{M}}$ is induced by $\widehat{\pi}$.

**Lemma B.7.** *Let $\widehat{\mathrm{M}}, \widehat{\mathrm{M}}_a$ and $\widehat{\mathrm{M}}_b$ be the empirical Markov chains defined in B.2, then the induced Markov chain from $\widehat{\pi}$ is exactly $\widehat{\mathrm{M}}$ and the induced Markov chains from $\widehat{\pi}_a$ and $\widehat{\pi}_b$ are exactly $\widehat{\mathrm{M}}_a$ and $\widehat{\mathrm{M}}_b$.*

*Proof.* Consider $\widehat{\mathrm{M}}_{(a,b),(a',b')} = \frac{\sum_{t=1}^T p_t(a',b')p_t(a,b)}{\sum_{t=1}^T p_t(a,b)}$. The transition probability induced by $\widehat{\pi}$ is exactly

$$\mathbb{P}[(a',b')|(a,b)] = \sum_{(f,g):(f(b),g(a))=(a',b')} \widehat{\pi}(f,g) = \sum_{(f,g):(f(b),g(a))=(a',b')} \prod_{i \in [|\mathcal{A}|]} \frac{\sum_t p_t(x_i)p_t(y_i)}{\sum_{t=1} p_t(x_i)}$$

$$= \sum_{(f,g):(f(b),g(a))=(a',b')} \frac{\sum_t p_t(a,b)p_t(a',b')}{\sum_{t=1} p_t(a,b)} \prod_{i \in [|\mathcal{A}|],(x_i,y_i) \neq ((a,b),(a',b'))} \frac{\sum_t p_t(x_i)p_t(y_i)}{\sum_{t=1} p_t(x_i)}$$

$$= \frac{\sum_t p_t(a,b)p_t(a',b')}{\sum_{t=1} p_t(a,b)} \sum_{(f,g):(f(b),g(a))=(a',b')} \prod_{i \in [|\mathcal{A}|],(x_i,y_i) \neq ((a,b),(a',b'))} \frac{\sum_t p_t(x_i)p_t(y_i)}{\sum_{t=1} p_t(x_i)}$$

$$= \frac{\sum_t p_t(a,b)p_t(a',b')}{\sum_{t=1} p_t(a,b)},$$

where the last equality holds, because for fixed $(f,g)$ with $x_i = (a_i, b_i)$ and $y_i = (f(b_i), g(a_i))$, the product $\prod_{i \in [|\mathcal{A}|],(x_i,y_i) \neq ((a,b),(a',b'))} \frac{\sum_t p_t(x_i)p_t(y_i)}{\sum_{t=1} p_t(x_i)}$ is exactly the conditional probability $\widehat{\pi}((f,g)|(f(b),g(a)) = (a',b'))$. The result for $\widehat{\pi}_a$ and $\widehat{\pi}_b$ is shown similarly. □

The last step of the proof is to show that any limit point $\pi$ of $(\widehat{\pi})_T$ is necessarily a policy equilibrium. This is done through an argument by contradiction. In particular we assume that a limit point $\pi$ is not a policy equilibrium. The limit point $\pi$ induces a Markov chain M, which we can show is the limit point of the corresponding subsequence of $(\widehat{M})_T$ by using lemma B.7. Since $\pi$ is not a policy equilibrium, no stationary distribution of M can satisfy the inequalities (3). We can now show that the subsequence of $(\widehat{\sigma})_T$ which are stationary distributions of the corresponding $\widehat{M}$'s, converges to a stationary distribution of M. This, however, is a contradiction because of the next theorem.

**Theorem B.8.** *Let $P$ be the set of all product distributions $\sigma \times \sigma_a \times \sigma_b$ which satisfy the inequalities in 3:*

$$\mathbb{E}_{(a,b)\sim\sigma}\left[u_1(a,b)\right] \geq \mathbb{E}_{(a,b)\sim\sigma_a}\left[u_1(a,b)\right]$$
$$\mathbb{E}_{(a,b)\sim\sigma}\left[u_2(a,b)\right] \geq \mathbb{E}_{(a,b)\sim\sigma_b}\left[u_2(a,b)\right].$$

*Let $\widehat{\sigma}^T$ be the empirical distribution of play after $T$ rounds and let $\widehat{\sigma}_a^T$ be the empirical distribution when player 1 switches to playing action $a$ and define $\widehat{\sigma}_b^T$ similarly for player 2. Then the product distribution $\widehat{\sigma}^T \times \widehat{\sigma}_a^T \times \widehat{\sigma}_b^T$ converges to weakly to the set $P$.*

*Proof.* Theorem B.8 follows from the fact that convergence in the Prokhorov metric implies weak convergence. First notice that by Prokhorov's theorem $\mathcal{P}(\mathcal{A})$ is a compact metric space with the Prokhorov metric. Thus by Tychonoff's Theorem the product space $\mathcal{P}(\mathcal{A})^3$ is compact in the maximum metric. Suppose for a contradiction that the sequence $(\widehat{\sigma}^T \times \widehat{\sigma}_a^T \times \widehat{\sigma}_b^T)_T$ does not converge to the set $S$. This implies that there exists some subsequence $(\widehat{\sigma}^k \times \widehat{\sigma}_a^k \times \widehat{\sigma}_b^k)_k$, converging to some $\widehat{\sigma} \times \widehat{\sigma}_a \times \widehat{\sigma}_b \notin S$. If $\widehat{\sigma} \times \widehat{\sigma}_a \times \widehat{\sigma}_b \notin S$, then either $\mathbb{E}_{(a,b)\sim\sigma}\left[u_1(a,b)\right] < \mathbb{E}_{(a,b)\sim\sigma_a}\left[u_1(a,b)\right]$ or $\mathbb{E}_{(a,b)\sim\sigma}\left[u_2(a,b)\right] < \mathbb{E}_{(a,b)\sim\sigma_b}\left[u_2(a,b)\right]$. WLOG suppose the first inequality holds. From our assumption, the continuity of $u_1$ and the definition of the maximum metric we have $\lim_{k\to\infty}\mathbb{E}_{(a,b)\sim\widehat{\sigma}^k}\left[u_1(a,b)\right] = \mathbb{E}_{(a,b)\sim\widehat{\sigma}}\left[u_1(a,b)\right]$ and $\lim_{k\to\infty}\mathbb{E}_{(a,b)\sim\widehat{\sigma}_a^k}\left[u_1(a,b)\right] = \mathbb{E}_{(a,b)\sim\widehat{\sigma}_a}\left[u_1(a,b)\right]$. Notice that by the fact $\widehat{\sigma}_a^k$ is the average empirical distribution if player 1 changed its play to the fixed action $a \in \mathcal{A}_1$ and $\widehat{\sigma}^k$ being the average empirical distribution it holds that $\mathbb{E}_{(a,b)\sim\widehat{\sigma}^k}\left[u_1(a,b)\right] - \mathbb{E}_{(a,b)\sim\widehat{\sigma}_a^k}\left[u_1(a,b)\right] \geq -o(1)$ and thus $\lim_{k\to\infty}\left[\mathbb{E}_{(a,b)\sim\widehat{\sigma}^k}\left[u_1(a,b)\right] - \mathbb{E}_{(a,b)\sim\widehat{\sigma}_a^k}\left[u_1(a,b)\right]\right] \geq 0$. The above implies:

$$0 \leq \lim_{k\to\infty}\left[\mathbb{E}_{(a,b)\sim\widehat{\sigma}^k}\left[u_1(a,b)\right] - \mathbb{E}_{(a,b)\sim\widehat{\sigma}_a^k}\left[u_1(a,b)\right]\right]$$
$$= \lim_{k\to\infty}\mathbb{E}_{(a,b)\sim\widehat{\sigma}^k}\left[u_1(a,b)\right] - \lim_{k\to\infty}\mathbb{E}_{(a,b)\sim\widehat{\sigma}_a^k}\left[u_1(a,b)\right]$$
$$= \mathbb{E}_{(a,b)\sim\widehat{\sigma}}\left[u_1(a,b)\right] - \mathbb{E}_{(a,b)\sim\widehat{\sigma}_a}\left[u_1(a,b)\right] < 0,$$

which is a contradiction. Since $\mathcal{A} \times \mathcal{A} \times \mathcal{A}$ is separable then convergence in the Prokhorov metric in $\mathcal{P}(\mathcal{A} \times \mathcal{A} \times \mathcal{A})$ is equivalent to weak convergence. Again we can argue by contradiction – if we assume that $(\widehat{\sigma}^T \times \widehat{\sigma}_a^T \times \widehat{\sigma}_b^T)_T$ doesn't converge to the set $S$ in the Prokhorov metric, then there exists some subsequence $(\widehat{\sigma}^k \times \widehat{\sigma}_a^k \times \widehat{\sigma}_b^k)_k$ which converges to some $\mu \in \mathcal{P}(\mathcal{A}^3)$ such that $\mu \notin S$. First we argue that $\mu$ must be a product measure i.e. $\mu = (\widehat{\sigma} \times \widehat{\sigma}_a \times \widehat{\sigma}_b)$. Let $(\widehat{\sigma}^j \times \widehat{\sigma}_a^j \times \widehat{\sigma}_b^j)_j$ be a convergence subsequence of $(\widehat{\sigma}^k \times \widehat{\sigma}_a^k \times \widehat{\sigma}_b^k)_k$ in $\mathcal{P}(\mathcal{A})^3$, with limit $(\widehat{\sigma} \times \widehat{\sigma}_a \times \widehat{\sigma}_b)$, then each of $\widehat{\sigma}^j$, $\widehat{\sigma}_a^j$ and $\widehat{\sigma}_b^j$ converge weakly to $\widehat{\sigma}$, $\widehat{\sigma}_a$ and $\widehat{\sigma}_b$ respectively and thus $(\widehat{\sigma}^j \times \widehat{\sigma}_a^j \times \widehat{\sigma}_b^j)_j$ converges weakly to $\widehat{\sigma} \times \widehat{\sigma}_a \times \widehat{\sigma}_b$ and thus it converges in the Prokhorov metric of $\mathcal{P}(\mathcal{A}^3)$. This implies that $(\widehat{\sigma}^k \times \widehat{\sigma}_a^k \times \widehat{\sigma}_b^k)_k$ also converges weakly to $\widehat{\sigma} \times \widehat{\sigma}_a \times \widehat{\sigma}_b$ and so $\mu$ is a product measure. Again since $\mu \notin S$, assume WLOG $\mathbb{E}_{(a,b)\sim\widehat{\sigma}}\left[u_1(a,b)\right] < \mathbb{E}_{(a,b)\sim\widehat{\sigma}_a}\left[u_1(a,b)\right]$. Define $f : \mathcal{A}^3 \to \mathbb{R}$, $f(a,b,c,d,e,f) = u_1(a,b) - u_1(c,d)$. $f$ is continuous and from the no-policy regret of the pair $\widehat{\sigma}^k, \widehat{\sigma}_a^k$ we have:

$$0 \leq \lim_{k\to\infty}\mathbb{E}_{(a,b,c,d,e,f)\sim(\widehat{\sigma}^k\times\widehat{\sigma}_a^k\times\widehat{\sigma}_b^k)_k}\left[f(a,b,c,d,e,f)\right] = \mathbb{E}_{(a,b,c,d,e,f)\sim\mu}\left[f(a,b,c,d,e,f)\right] < 0,$$

which is again a contradiction. $\qquad\square$

For completeness we restate the main result below and give its proof in full.

**Theorem B.9.** *If the algorithms played by player 1 in the form of $f_t$ and player 2 in the form of $g_t$ give sub-linear policy regret sequences, then sequence of product distributions $(\widehat{\sigma}^T \times \widehat{\sigma}_a^T \times \widehat{\sigma}_b^T)_{T=1}^{\infty}$ converges weakly to the set $S$.*

*Proof of Theorem 4.8.* We consider the sequence of empirical distributions $\widehat{\pi}^T$ defined in B.5, over the functional space $\mathcal{F}_1 \times \mathcal{F}_2$ and show that this sequence must converge to the set of all policy equilibria $\Pi$ in the Prokhorov metric on $\mathcal{P}(\mathcal{F}_1 \times \mathcal{F}_2)$. First, notice that since the functions $f : \mathcal{A}_2 \to \mathcal{A}_1$ are from finite sets of actions to finite sets of actions, we can consider the set $\mathcal{F}_1$ as a subset of a finite dimensional vector space, with the underlying field of real numbers and the metric induced by the $l_1$ norm. Similarly, we can also equip $\mathcal{F}_2$ with the $l_1$ norm. Since both $\mathcal{F}_1$ and $\mathcal{F}_2$ are closed sets with respect to this metric and they are clearly bounded, they are compact. Thus the set $\mathcal{F}_1 \times \mathcal{F}_2$ is a compact set with the underlying metric $d$ being the maximum metric. By Prokhorov's theorem we know that $\mathcal{P}(\mathcal{F}_1 \times \mathcal{F}_2)$ is a compact metric space with the Prokhorov metric. Suppose that the sequence $(\widehat{\pi}^T)_T$ does not converge to $\Pi$. This implies that there is some convergent subsequence $(\widehat{\pi}^t)_t$ with a limit $\pi$ outside of $\Pi$. Let $M$ be the Markov chain induced by $\pi$ and let $\widehat{M}^T$ be the Markov chain induced by $\widehat{\pi}^T$.

First we show that $\lim_{t\to\infty} \|\widehat{M}^t - M\|_1 = 0$. Recall that $M_{(a,b),(a',b')} = \sum_{(f,g):f(b)=a',g(a)=b'} \pi(f,g)$ and that by lemma B.7 $\widehat{M}^t_{(a,b),(a',b')} = \sum_{(f,g):f(b)=a',g(a)=b'} \widehat{\pi}^t(f,g)$. Notice that $f,g$ are continuous functions on $\mathcal{F}_1$ and $\mathcal{F}_2$, since the topology induced by the $l_1$ metric on both sets is exactly the the discrete topology and every function from a topological space equipped with the discrete topology is continuous. Since convergence in the Prokhorov metric implies weak convergence, we have that for any fixed $f,g$, $\lim_{t\to\infty} \widehat{\pi}^t(f,g) = \pi(f,g)$. Since the sum $\sum_{(f,g):f(b)=a',g(a)=b'} \widehat{\pi}^t(f,g)$ is finite this implies that $\lim_{t\to\infty} \sum_{(f,g):f(b)=a',g(a)=b'} \widehat{\pi}^t(f,g) = \sum_{(f,g):f(b)=a',g(a)=b'} \pi(f,g)$ and so $\lim_{t\to\infty} \|\widehat{M}^t - M\|_1 = 0$.

Next we show that any convergent subsequence $(\widehat{\sigma}^k)_k$ of $(\widehat{\sigma}^t)_t$ in the Prokhorov metric, converges to a stationary distribution $\sigma$ of $M$. First notice that $(\widehat{\sigma}^k)_k$ exists, since $\mathcal{P}(\mathcal{A})$ is compact. Next, suppose $\sigma$ is the limit of $(\widehat{\sigma}^k)_k$ in the Prokhorov metric. This implies that $\lim_{k\to\infty} \widehat{\sigma}^k(a,b) = \sigma(a,b)$, in particular if we consider $\mathcal{A} \subset \mathbb{R}^{|\mathcal{A}|}$ and $\widehat{\sigma}^k, \sigma \in \mathbb{R}^{|\mathcal{A}|}$ as vectors, then the above implies that $\lim_{k\to\infty} \|\sigma - \widehat{\sigma}^k\|_1 = 0$. Next we construct the following sequence $(\sigma_{k_n})_{k_n}$ of stationary distributions of $M$ – choose $k_n$ large enough, so that $\|M - \widehat{M}^{k_n}\| \le \frac{1}{n}$. Such a $k_n$ exists, because $(\widehat{M}^k)_k$ is a subsequence of $(\widehat{M}^t)_t$ which converges to $M$. By lemma B.3, there exists a stationary distribution $\sigma_{k_n}$ of $M$ such that $\|\widehat{\sigma}_{k_n} - \sigma_{k_n}\|_1 \le \frac{c}{n}$, for some constant $c$. We show that $\sigma_{k_n}$ converges to $\sigma$. Fix some $\epsilon > 0$, we find an $N$, such that for any $n \ge N$ we have $\|\sigma_{k_n} - \sigma\|_1 < \epsilon$. Notice that $\|\sigma_{k_n} - \sigma\|_1 \le \|\sigma_{k_n} - \widehat{\sigma}_{k_n}\|_1 + \|\widehat{\sigma}_{k_n} - \sigma\|_1$. Since $\|\sigma_{k_n} - \widehat{\sigma}_{k_n}\|_1 \le \frac{c}{n}$ and by convergence, we know that for $\frac{\epsilon}{2}$, there exists $N'$ such that for any $n \ge N'$, $\|\widehat{\sigma}_{k_n} - \sigma\|_1 < \frac{\epsilon}{2}$, we can set $N = \max\left(\frac{2}{c\epsilon}, N_1\right)$. Suppose, for a contradiction, that $\sigma$ is not a stationary distribution of $M$. Then there exists some $\epsilon$ such that $\|\sigma^\top M - \sigma\|_2 > \epsilon$. This implies:

$$\epsilon < \|\sigma^\top M - \sigma\|_2 \le \|\sigma^\top M - \sigma_{k_n}^\top M\|_2 + \|\sigma_{k_n}^\top M - \sigma\|_2 < 2\|\sigma_{k_n} - \sigma\|_2,$$

where the last inequality holds by the fact $\sigma - \sigma_{k_n}$ is not a stationary distribution of $M$ and thus $M$ can only shrink the difference as a stochastic matrix. The inequality $2\|\sigma_{k_n} - \sigma\|_2 > \epsilon$ is a contradiction since we know that $\sigma_{k_n}$ converges to $\sigma$ and thus $\sigma$ is a stationary distribution of $M$. Since strong convergence, implies weak convergence, which in hand implies convergence in the Prokhorov metric for separable metric spaces, we have shown that every convergent subsequence of $(\widehat{\sigma}_t)_t$ converges to a stationary distribution of $M$ in the Prokhorov metric.

Next, we show that $(\widehat{\pi}^t_a)_t$ converges to $\pi_a$. By assumption $(\widehat{\pi}^t)_t$ converges weakly to $\pi$. Since we are are in a finite dimensional space, we also have strong convergence. In particular, for any $g \in \mathcal{F}_2$, we have

$$\lim_{t\to\infty} \sum_{f\in\mathcal{F}_1} \widehat{\pi}^t(f,g) = \sum_{f\in\mathcal{F}_1} \lim_{t\to\infty} \widehat{\pi}^t(f,g) = \sum_{f\in\mathcal{F}_1} \pi(f,g)$$

and so the sequence of marginal distribution also converges to the respective marginal of $\pi$. Since $\widehat{\pi}^t_a$ is exactly the product distribution of the dirac distribution over $\mathcal{F}_1$ putting all weight on the constant function mapping everything to the fixed action $a$ and the marginal of $\widehat{\pi}^t$ over $\mathcal{F}_2$, by the convergence of marginals we conclude that $(\widehat{\pi}^t_a)_t$ converges to $\pi_a$ in the strong sense and thus in the Prokhorov metric. In the same way we can show that $(\widehat{\pi}^t_b)_t$ converges to $\pi_b$.

With a similar argument as for $(\widehat{\sigma}^t)$ we show that every convergent subsequence of $(\widehat{\sigma}^t_a)_t$ converges to a stationary distribution of $M_a$ and any convergent subsequence of $(\widehat{\sigma}^t_b)_t$ converges to a stationary

distribution of $M_b$. Because of the construction in theorem B.1 and the convergence of $\widehat{\pi}_a$ to $\pi_a$, we can guarantee that $(\widehat{\sigma}_a^t)_t$ converges precisely to $\sigma_a$:

$$\sigma_a(a, b) = \sum_{f, g: g(a) = b} \pi_a(f, g) = \sum_{f, g: g(a) = b} \lim_{t \to \infty} \widehat{\pi}_a^t(f, g)$$

$$= \lim_{t \to \infty} \sum_{f, g: g(a) = b} \widehat{\pi}_a^t(f, g) = \lim_{t \to \infty} \widehat{\sigma}_a^t(a, b).$$

Similarly $\widehat{\sigma}_b^t$ converges to $\widehat{\sigma}_b$. However, we assumed that $\pi$ is not a policy equilibrium and thus no stationary distributions of $M$, $M_a$ and $M_b$ can satisfy the policy equilibrium inequalities. We now arrive at a contradiction since by theorem B.8 and the above, we have that any limit point of $(\widehat{\sigma}^t)_t$ and the corresponding distributions for fixed actions $a$ and $b$ are stationary distributions of $M$, $M_a$ and $M_b$, respectively, which satisfy the policy equilibrium inequalities. $\square$

Not surprisingly, we are able to show that the empirical distributions of observed play, also converge to the set of $S$ almost surely.

**Corollary B.10.** *The sequence of product distributions $(\tilde{\sigma} \times \tilde{\sigma}_a \times \tilde{\sigma}_b)_{T=1}^\infty$ converges weakly to the set $S$ almost surely.*

*Proof.* We are going to show that for any convergent subsequence $(\widehat{\sigma}^t)_t$ of $(\widehat{\sigma})_T$ with limit point $\sigma$, the corresponding subsequence $(\tilde{\sigma}^t)_t$ converges to $\sigma$ a.s.. Let $E$ be the event that $\lim_{t \to \infty} \|\sigma - \tilde{\sigma}^t\|_1 = 0$. Consider the following event $\lim_{t \to \infty} \|\sigma - \widehat{\sigma}^t\|_1 + \lim_{t \to \infty} \|\widehat{\sigma}^t - \tilde{\sigma}^t\|_1 = 0$ denoted by $E'$. Notice that every time $E'$ occurs $E$ also occurs because

$$\|\sigma - \tilde{\sigma}^t\|_1 \leq \|\sigma - \widehat{\sigma}^t\|_1 + \|\widehat{\sigma}^t - \tilde{\sigma}^t\|_1,$$

and taking limits, preserves the non-strict inequality. By Theorem B.19, $E'$ occurs with probability 1 and thus $E$ occurs w.p. 1 as well. $\square$

## B.2 Proof that CCEs are a subset of policy equilibria

**Lemma B.11.** *For any fixed game $\mathcal{G}$ and a CCE $\sigma$ of $\mathcal{G}$, there exist on average $(m, R(T))$ stable no-external regret algorithms, which if the players follow, the empirical distribution of play, converges to $\sigma$. Here $m < o(T)$ and $R(T) \leq O(\sqrt{T})$.*

*Proof.* Fix the time horizon to be $T$. We assume that at round $t$ of the game, player $i$ has access to an oracle which provides $\widehat{\sigma}_t$, the empirical distribution of play. The algorithm of player $i$ is the following – split the time horizon into mini-batches of size $\sqrt{T}$. For the first mini-batch the player plays according to $\sigma$. At the end of the mini-batch, the oracle provides the player with $\widehat{\sigma}_t$ and the player checks if $\|\widehat{\sigma}_t - \sigma\|_1 \leq \frac{|\mathcal{A}|}{T^{1/6}}$. If the condition is satisfied, then the algorithm continues playing according to $\sigma$. At the $j$-th epoch, the player checks if $\|\widehat{\sigma}_t - \sigma\|_1 \leq \frac{|\mathcal{A}|}{jT^{1/6}}$. If at any of the epochs the inequality turns out to be false, then the player switches to playing Exp3 at the end of the next epoch for the rest of the game i.e. if at the end of the $j$-th epoch, the inequality does not hold, then at the end of the $j + 1$-st epoch, the player switches to Exp3. First we show that this is a no-external regret algorithm. Suppose until epoch $j$ the player has not switched to Exp3 and then at epoch $j$ the player switches. Let $u_t$ denote the utility function which the player observed at time $t$. Let $(a_t)_{t=1}^T$ be the action sequence of the player. For any fixed action $a \in \mathcal{A}_i$ we have:

$$\mathbb{E}\left[\sum_{t=1}^T u_t(a) - \sum_{t=1}^T u_t(a_t)\right] = \mathbb{E}\left[\sum_{t=1}^{(j-1)\sqrt{T}} u_t(a) - \sum_{t=1}^{(j-1)\sqrt{T}} u_t(a_t)\right] + \sum_{t=(j-1)\sqrt{T}+1}^{(j+1)\sqrt{T}} [u_t(a) - u(a_t)]$$

$$+ \mathbb{E}\left[\sum_{t=(j+1)\sqrt{T}+1}^T u_t(a) - u(a_t)\right]$$

$$\leq \mathbb{E}\left[\sum_{t=1}^{(j-1)\sqrt{T}} u_t(a) - \sum_{t=1}^{(j-1)\sqrt{T}} u_t(a_t)\right] + 2\sqrt{T} + c\sqrt{|\mathcal{A}_i| \log(|\mathcal{A}_i|) T}$$

where in the first inequality we have used the fact that the utilities are in $[0,1]$ and the bound on the regret of Exp3 where $c$ is some constant. We now bound the term $\mathbb{E}\left[\sum_{t=1}^{(j-1)\sqrt{T}} u_t(a) - \sum_{t=1}^{(j-1)\sqrt{T}} u_t(a_t)\right]$ via two applications of lemma B.13.

$$\frac{1}{T}\mathbb{E}\left[\sum_{t=1}^{(j-1)\sqrt{T}} u_t(a) - \sum_{t=1}^{(j-1)\sqrt{T}} u_t(a_t)\right] = \mathbb{E}_{(a',b')\sim\widehat{\sigma}_t}\left[u_i(a,b')\right] - \mathbb{E}_{(a',b')\sim\widehat{\sigma}^t}\left[u_i(a',b')\right]$$

$$= \mathbb{E}_{(a',b')\sim\widehat{\sigma}_t}\left[u_i(a,b')\right] - \mathbb{E}_{(a',b')\sim\sigma}\left[u_i(a,b')\right] + \mathbb{E}_{(a',b')\sim\sigma}\left[u_i(a,b')\right] - \mathbb{E}_{(a',b')\sim\sigma}\left[u_i(a',b')\right]$$

$$+ \mathbb{E}_{(a',b')\sim\sigma}\left[u_i(a',b')\right] - \mathbb{E}_{(a',b')\sim\widehat{\sigma}_t}\left[u_i(a',b')\right]$$

$$\leq \|\mathrm{P}_i\|\frac{(j-1)|\mathcal{A}|}{jT^{1/6}} + \langle \mathrm{P}_i, \mathbb{E}_{(a',b')\sim\sigma}\left[b'a'^{\top}\right]\rangle - \langle \mathrm{P}_i, \mathbb{E}_{(a',b')\sim\widehat{\sigma}_t}\left[b'a'^{\top}\right]\rangle$$

$$\leq 2\|\mathrm{P}_i\|\frac{|\mathcal{A}|}{T^{1/6}},$$

where the first inequality follows from lemma B.13 and the fact $\sigma$ is a CCE. The second inequality again follows from lemma B.13. The above implies $\mathbb{E}\left[\sum_{t=1}^{(j-1)\sqrt{T}} u_t(a) - \sum_{t=1}^{(j-1)\sqrt{T}} u_t(a_t)\right] \leq \frac{\sqrt{T}|\mathcal{A}|^2}{T^{1/6}}$. This follows fact that the spectral norm of the utility matrix for player $i$ can not exceed $|\mathcal{A}|$, which again follows from the boundedness of utilities.

Next, we show that if all players, play according to the algorithm, then with high probability, the empirical distribution of play converges to $\sigma$. For this, we analyze the probability of the event that $\|\widehat{\sigma}_t - \sigma\|_1 > \frac{|\mathcal{A}|}{T^{1/6}}$ after the first epoch. By Azuma's inequality, we have that $\mathbb{P}\left[|\widehat{\sigma}_t(x) - \sigma(x)| > \frac{\epsilon}{|\mathcal{A}|}\right] \leq 2\exp\left(-\epsilon^2 t/(2|\mathcal{A}|)\right)$, for any fixed action $x \in \mathcal{A}$. By a union bound over all possible actions, we get that $\mathbb{P}\left[\|\widehat{\sigma}_1 - \sigma\|_1 > \epsilon\right] \leq 2|\mathcal{A}|\exp\left(-\epsilon^2 t/(2|\mathcal{A}|)\right)$. Plugging in $\epsilon = \frac{|\mathcal{A}|}{T^{1/6}}$, we get that the probability the algorithm fails after the first epoch is bounded by $2|\mathcal{A}|\exp\left(-T^{1/6}|\mathcal{A}|/2\right)$. Now the probability that the algorithm fails after epoch $j$ is bounded by

$$\left(1 - 2|\mathcal{A}|\exp\left(-T^{1/6}|\mathcal{A}|/2\right)\right)^{j-1} 2|\mathcal{A}|\exp\left(-j^2 T^{1/6}|\mathcal{A}|/2\right) < 2|\mathcal{A}|\exp\left(-T^{1/6}|\mathcal{A}|/2\right).$$

A very pessimistic union bound now gives us that the probability the algorithm fails to converge to $\sigma$, provided that all players use it, is upper bounded by $2\sqrt{T}|\mathcal{A}|\exp\left(-T^{1/6}|\mathcal{A}|/2\right)$.

Using a doubling trick, we can achieve the same results for any $T$. As in theorem B.19, we can now get $\limsup_{T\to\infty} \|\widehat{\sigma}_T - \sigma\|_1 = 0$ almost surely and thus the empirical distribution of play converges to $\sigma$ a.s.. To check the stability of the algorithm we consider three possibilities – the player is playing according to $\sigma$ and is not in an epoch after which they switch to Exp3, the player is playing according to $\sigma$ and is in an epoch after which they switch to Exp3 and the player has switched to playing Exp3. We take the view of player 1. Consider the first case and suppose the current epoch of the algorithm is $j$. In particular consider time step $j\sqrt{T} + i$ at which the play was according to $\sigma_{j\sqrt{T}+i} = \sigma$ and suppose player 2 in the last $m < o(\sqrt{T})$ switched from playing distributions $(p_t)_{t=j\sqrt{T}+i-m}^{j\sqrt{T}}$ to $(\tilde{p}_t)_{t=j\sqrt{T}+i-m}^{j\sqrt{T}}$ so that player 1 plays $\tilde{\sigma}_{j\sqrt{T}+i}$. Since $m < o(\sqrt{T})$, the only change in the current algorithm can occur at the $j-1$-st epoch and is in the form of the algorithm decides to switch to Exp3. Since the switch can only occur after the $j$-th epoch is done, this implies that at time $j\sqrt{T} + i$ again the play is according to the distribution $\sigma$ and we have $\|\sigma_{j\sqrt{T}+i} - \tilde{\sigma}_{j\sqrt{T}+i}\|_1 = 0$. Consider the second case in which at the end of epoch $j$ the algorithm switches to playing Exp3. If the past $m$ actions of player 2 change, this can only change the decision at the end of $j-1$-st epoch of player 1 to switch to Exp3. However, as in the previous case, the player at the $j$-th epoch is still playing according to $\sigma$ and again the algorithm is stable at each iteration during the epoch. Finally, in the third case we have stability from the fact Exp3 is stable and once we enter an epoch during which Exp3 is played, changing the past $m$-actions of player 2 can not change the decision of player 1 to switch to Exp3, since $m < o(\sqrt{T})$ and the epochs are of size $\sqrt{T}$.

$\square$

As a consequence of the proof we see that, the algorithm is on average stable for any $m < o(\sqrt{T})$.

Table 1: Utility for player 1

| player 1\player 2 | c_1 | c_2 |
|---|---|---|
| **a** | 3/4 | 0 |
| **b** | 1 | 0 |

Theorems 4.9 and 4.10 are combined in the the following theorem.

**Theorem B.12.** *For any CCE $\sigma$ of a 2-player game $\mathcal{G}$, there exists a policy-equilibrium $\pi$, which induces a Markov chain M, with stationary distribution $\sigma$. Further, there exists a 2-player game $\mathcal{G}$ and product distributions $\sigma \times \sigma_a \times \sigma_b \in S$, where $S$ is defined in 4.6, such that $\sigma$ is not a CCE of $\mathcal{G}$.*

*Proof.* First note that lemma B.11 guarantees that for any CCE $\sigma$ there exists an $m$-stable on average no-external regret algorithm, which generates an empirical distribution of play $\tilde{\sigma}$ converging to $\sigma$ almost surely. From the $m$-stability of the algorithm and theorem A.2, we know that if both players play according to $\tilde{\sigma}$ then the empirical sequence of play also has no-policy regret and thus by theorem B.9, we know that $\tilde{\sigma} \times \tilde{\sigma}_a \times \tilde{\sigma}_b$ converges to the set of policy-regret equilibria.

To show the second part of the theorem, we consider a game in which the pay-off matrix for player 2 is just some constant multiple of $\mathbf{1}_{\mathcal{A}_1} \mathbf{1}_{\mathcal{A}_2}^\top$, where $\mathbf{1}_{\mathcal{A}_i} \in \mathbb{R}^{|\mathcal{A}_i|}$ is the vector with each entry equal to 1. This implies that whatever sequence player 2 chooses to play, they will have no-policy regret and no-external regret, since the observed utility in expectation is going to be equal for all actions $b \in \mathcal{A}_2$. Let $\mathcal{A}_1 = \{a, b\}$. The utility matrix of player 1 is given in 1. The strategy of player 2 is the following:

$$f_t(a_{t-1}, a_t) = \begin{cases} 1 & a_{t-1} = a, a_t = b \\ \frac{3}{4} & a_{t-1} = a_t = a \\ 0 & \text{otherwise} \end{cases}.$$

Similarly to the proof of theorem A.1, we can show that any no-policy regret strategy for player 1, incurs linear external regret. For any time step $t$, let $\widehat{\sigma}_t^1$ be the dirac distribution for player 1, putting all probability mass on action $a$ and let $\widehat{\sigma}_t^2$ be the dirac distribution putting all probability mass on the action played by player 2 at time $t$. Let $\widehat{\sigma} = \frac{1}{T} \sum_{t=1}^T \widehat{\sigma}_t^1 (\widehat{\sigma}_t^2)^\top$ be the empirical distribution after $T$ rounds. Then by theorem B.9 $\widehat{\sigma} \times \widehat{\sigma}_a \times \widehat{\sigma}_b$ converges to the set of policy equilibria. However, by construction, if player 1, plays according to $\widehat{\sigma}$, then they will have linear external regret and thus the $\widehat{\sigma}$ can not be converging to the set of CCEs. $\qquad\square$

### B.3 Auxiliary results.

**Lemma B.13.** *Let $\sigma$ and $\sigma'$ be two distributions supported on a finite set and let $f$ be a utility/loss function uniformly bounded by 1. If $\|\sigma - \sigma'\|_1 \le \epsilon$ then $|\mathbb{E}_{a \sim \sigma}[f(a)] - \mathbb{E}_{a \sim \sigma'}[f(a)]| \le \epsilon$.*

*Proof.*

$$|\mathbb{E}_{s \sim \sigma}[f(s)] - \mathbb{E}_{s' \sim \sigma'}[f(s)]| = |\sum_{s \in S} \sigma(s)f(s) - \sum_{s \in S} \sigma'(s)f(s)|$$

$$= |\sum_{s \in S} f(s)(\sigma(s) - \sigma'(s))| \le \sum_{s \in S} |\sigma(s) - \sigma'(s)| = \|\sigma - \sigma'\|_1 \le \epsilon.$$

$\square$

**Lemma B.14.** *Let $M \in \mathbb{R}^{d \times d}$ and $\widehat{M} \in \mathbb{R}^{d \times d}$ be two row-stochastic matrices, such that $\|M - \widehat{M}\| \le \epsilon$, then for any stationary distribution $\widehat{\sigma}$ of $\widehat{M}$, there exists a stationary distribution $\sigma$ of M, such that $\|\widehat{\sigma} - \sigma\|_1 \le \frac{4d^2 \epsilon}{\delta}$.*

*Proof.* Let $U \in \mathbb{R}^{d \times k}$ be the left singular vectors corresponding to the singular value 1 of M and let $\widehat{U} \in \mathbb{R}^{d \times l}$ be the left singular vectors corresponding to the singular value 1 of $\widehat{M}$. First notice that

$$\|MM^\top - \widehat{M}\widehat{M}^\top\| \le \|MM^\top - M\widehat{M}^\top\| + \|M\widehat{M}^\top - \widehat{M}\widehat{M}^\top\| \le (\|M\| + \|\widehat{M}\|)\|M - \widehat{M}\| \le 2\epsilon$$

Denote the eigen-gap of $M$ by $\delta$, then by Wedin's theorem (see for example lemma B.3 in Allen-Zhu and Li (2016)) theorem we have

$$\|\widehat{U}^\top U^\perp\| \le \frac{\|MM^\top - \widehat{M}\widehat{M}^\top\|}{\delta} \le \frac{2\epsilon}{\delta}.$$

WLOG assume $\widehat{\sigma} = \frac{(\widehat{U})_i}{\|(\widehat{U})_i\|_1}$. This implies that $\|\widehat{\sigma}^\top U^\perp\|_2 \le \frac{2d\epsilon}{\delta}$ and thus:

$$\|UU^\top \widehat{\sigma} - \widehat{\sigma}\|_2 = \|(I - UU^\top)\widehat{\sigma}\|_2 = \|U^\perp (U^\perp)^\top \widehat{\sigma}\|_2 \le \|\widehat{\sigma}^\top U^\perp\|_2 \le \frac{2d\epsilon}{\delta}.$$

Let $\sigma_i = \frac{U_i}{\|U_i\|_1}$ be the stationary distribution of $M$, corresponding to the $i$-th left singular vector and let $\alpha_i = (U^\top \widehat{\sigma})_i \|U_i\|_1 \ge 0$. Then we have $\|\sum_i \alpha_i \sigma_i - \widehat{\sigma}\|_1 \le \frac{2d^2\epsilon}{\delta}$, where the inequality follows from the derivation above and the inequality between $l_1$ and $l_2$ norms. Let $\sigma = \frac{\sum_i \alpha_i \sigma_i}{\|\sum_i \alpha_i \sigma_i\|_1}$. This is a stationary distribution of $M$, since

$$\sigma^\top M = \frac{1}{\|\sum_i \alpha_i \sigma_i\|_1} \sum_i \alpha_i \sigma_i^\top M = \frac{\sum_i \alpha_i \sigma_i}{\|\sum_i \alpha_i \sigma_i\|_1} = \sigma.$$

Notice that by reverse triangle inequality we have

$$\left| \|\sum_i \alpha_i \sigma_i\|_1 - \|\widehat{\sigma}\|_1 \right| \le \frac{2d^2\epsilon}{\delta},$$

or equivalently

$$\left| \|\sum_i \alpha_i \sigma_i\|_1 - 1 \right| \le \frac{2d^2\epsilon}{\delta}.$$

Thus we have:

$$\|\sigma - \widehat{\sigma}\|_1 \le \|\sum_i \alpha_i \sigma_i - \widehat{\sigma}\|_1 + \|\sigma - \sum_i \alpha_i \sigma_i - \widehat{\sigma}\|_1 \le \frac{2d^2\epsilon}{\delta} + \|\sigma\|_1 \left| 1 - \|\sum_i \alpha_i \sigma_i\|_1 \right| \le \frac{4d^2\epsilon}{\delta}.$$

$\square$

**Corollary B.15.** *Let the empirical distribution of observed play be $\tilde{\sigma}^T = \frac{1}{T}\sum_{t=1}^T \delta_t$, the empirical distribution of play if player 1 deviated to playing fixed action $a \in \mathcal{A}_1$ be $\tilde{\sigma}_a^T$ and the empirical distribution of play if player 2 to action $b \in \mathcal{A}_2$ be $\tilde{\sigma}_b^T$. The sequence $(\tilde{\sigma}^T, \tilde{\sigma}_a^T, \tilde{\sigma}_b^T)_T$ converges to the set $P$ almost surely.*

*Proof.* Lemma B.13, together with theorem B.19 imply that

$$\limsup_{T\to\infty} |\mathbb{E}_{(a,b)\sim\tilde{\sigma}^T}[u_1(a,b)] - \mathbb{E}_{(a,b)\sim\widehat{\sigma}^T}[u_1(a,b)]| = 0$$

almost surely i.e.

$$\mathbb{P}\left[\limsup_{T\to\infty} |\mathbb{E}_{(a,b)\sim\tilde{\sigma}^T}[u_1(a,b)] - \mathbb{E}_{(a,b)\sim\widehat{\sigma}^T}[u_1(a,b)]| = 0\right] = 1.$$

Since

$$\limsup_{T\to\infty} |\mathbb{E}_{(a,b)\sim\tilde{\sigma}^T}[u_1(a,b)] - \mathbb{E}_{(a,b)\sim\widehat{\sigma}^T}[u_1(a,b)]| \ge \liminf_{T\to\infty} |\mathbb{E}_{(a,b)\sim\tilde{\sigma}^T}[u_1(a,b)] - \mathbb{E}_{(a,b)\sim\widehat{\sigma}^T}[u_1(a,b)]| \ge 0,$$

this implies that

$$\mathbb{P}\left[\liminf_{T\to\infty} |\mathbb{E}_{(a,b)\sim\tilde{\sigma}^T}[u_1(a,b)] - \mathbb{E}_{(a,b)\sim\widehat{\sigma}^T}[u_1(a,b)]| = 0\right] = 1.$$

On the other hand this implies

$$\mathbb{P}\left[\liminf_{T\to\infty} |\mathbb{E}_{(a,b)\sim\tilde{\sigma}^T}[u_1(a,b)] - \mathbb{E}_{(a,b)\sim\widehat{\sigma}^T}[u_1(a,b)]| = 0 \bigcap \limsup_{T\to\infty} |\mathbb{E}_{(a,b)\sim\tilde{\sigma}^T}[u_1(a,b)] - \mathbb{E}_{(a,b)\sim\widehat{\sigma}^T}[u_1(a,b)]| = 0\right]$$

and so

$$\mathbb{P}\left[\lim_{T\to\infty}|\mathbb{E}_{(a,b)\sim\tilde{\sigma}^T}\left[u_1(a,b)\right]-\mathbb{E}_{(a,b)\sim\hat{\sigma}^T}\left[u_1(a,b)\right]|=0\right]=1.$$

In a similar way we can get $\lim_{T\to\infty}|\mathbb{E}_{(a,b)\sim\tilde{\sigma}_a^T}\left[u_1(a,b)\right]-\mathbb{E}_{(a,b)\sim\hat{\sigma}_a^T}\left[u_1(a,b)\right]|=0$ a.s. The above imply that $\lim_{T\to\infty}\mathbb{E}_{(a,b)\sim\tilde{\sigma}^T}\left[u_1(a,b)\right]-\mathbb{E}_{(a,b)\sim\hat{\sigma}^T}\left[u_1(a,b)\right]=0$ a.s. and $\lim_{T\to\infty}\mathbb{E}_{(a,b)\sim\hat{\sigma}_a^T}\left[u_1(a,b)\right]-\mathbb{E}_{(a,b)\sim\tilde{\sigma}_a^T}\left[u_1(a,b)\right]=0$ a.s. and thus:

$$0=\lim_{T\to\infty}\mathbb{E}_{(a,b)\sim\tilde{\sigma}^T}\left[u_1(a,b)\right]-\mathbb{E}_{(a,b)\sim\hat{\sigma}^T}\left[u_1(a,b)\right]+\lim_{T\to\infty}\mathbb{E}_{(a,b)\sim\hat{\sigma}_a^T}\left[u_1(a,b)\right]-\mathbb{E}_{(a,b)\sim\tilde{\sigma}_a^T}\left[u_1(a,b)\right]$$

$$=\lim_{T\to\infty}\mathbb{E}_{(a,b)\sim\tilde{\sigma}^T}\left[u_1(a,b)\right]-\mathbb{E}_{(a,b)\sim\tilde{\sigma}_a^T}\left[u_1(a,b)\right]+\lim_{T\to\infty}\mathbb{E}_{(a,b)\sim\hat{\sigma}_a^T}\left[u_1(a,b)\right]-\mathbb{E}_{(a,b)\sim\hat{\sigma}^T}\left[u_1(a,b)\right]$$

a.s.. Since $\mathbb{E}_{(a,b)\sim\hat{\sigma}_a^T}\left[u_1(a,b)\right]-\mathbb{E}_{(a,b)\sim\hat{\sigma}^T}\left[u_1(a,b)\right]<o(1)$, this implies that

$$0\le-\lim_{T\to\infty}\mathbb{E}_{(a,b)\sim\hat{\sigma}_a^T}\left[u_1(a,b)\right]-\mathbb{E}_{(a,b)\sim\hat{\sigma}^T}\left[u_1(a,b)\right]$$

$$=\lim_{T\to\infty}\mathbb{E}_{(a,b)\sim\tilde{\sigma}^T}\left[u_1(a,b)\right]-\mathbb{E}_{(a,b)\sim\tilde{\sigma}_a^T}\left[u_1(a,b)\right]$$

a.s.. Now we can proceed as in the proof of theorem B.8. □

## B.4   Concentration of $\tilde{\mathrm{M}}$

**Lemma B.16.** *With probability at least* $1-|\mathcal{A}|6\exp\left(-\frac{T\epsilon^2}{4}\right)$ *it holds that* $|\frac{1}{T}\sum_{t=1}^T p_{t-1}(x_i)p_t(x_j)-\frac{1}{T}\sum_{t=1}^T \delta_{t-1}(x_i)\delta_t(x_j)|<\epsilon$ *and* $|\frac{1}{T}\sum_{t=1}^T p_t(x_i)-\frac{1}{T}\sum_{t=1}^T \delta_t(x_i)|<\epsilon$, *simultaneously for all* $i$.

*Proof.* We consider the random variable $Z_t=\delta_t(x_i)-p_t(x_i)$, notice that $\mathbb{E}\left[Z_t|p_1,\cdots,p_{t-1}\right]=0$ so that $\{Z_t\}_t$ is a bounded martingale sequence with $|Z_t|<1$ and thus by Azuma's inequality we have $\mathbb{P}\left[|\frac{1}{T}\sum_{t=1}^T Z_t|\ge\epsilon\right]<2\exp\left(-\frac{T\epsilon^2}{2}\right)$ which shows that $\frac{1}{T}\sum_{t=1}^T \delta_t(a_i)$ concentrates around $\frac{1}{T}\sum_{t=1}^T p_t(x_i)$. Let $R_t=\delta_{t-1}(x_i)\delta_t(x_j)-p_{t-1}(x_i)p_t(x_j)$ and consider the filtration $\{\mathcal{F}_t\}_t$, where $\mathcal{F}_1=\varnothing$, $\mathcal{F}_t=\Sigma(\delta_1,\cdots,\delta_t)$ is the sigma algebra generated by the random variables $\delta_1$ to $\delta_t$. Then $|R_{2t}|\le 1$ and $\mathbb{E}\left[R_{2t}|\mathcal{F}_1,\cdots,\mathcal{F}_{2t-2}\right]=0$, so $\{R_{2t}\}_t$ is also a bounded martingale difference and thus $\mathbb{P}\left[|\frac{1}{T}\sum_{t=1}^{\frac{T}{2}} R_{2t}|\ge\frac{\epsilon}{2}\right]<2\exp\left(-\frac{T\epsilon^2}{4}\right)$. A similar argument allows us to bound the sum of the $R_{2t+1}$'s and a union bound gives us $\mathbb{P}\left[|\frac{1}{T}\sum_{t=1}^T R_t|\ge\epsilon\right]<4\exp\left(-\frac{T\epsilon^2}{4}\right)$. A union bound over all $i$ finishes the proof. □

**Definition B.17.** Define the perturbed distribution of player $i$ at time $t$ to be $\tilde{p}_t^i=(1-\sqrt{|\mathcal{A}|\tilde{\epsilon}})p_t^i+\mathbf{1}\frac{\sqrt{|\mathcal{A}|\tilde{\epsilon}}}{|\mathcal{A}_i|}$.

**Lemma B.18.** *The difference of expected utilities from playing according to* $(\tilde{p}_t^i)_{t=1}^T$ *instead of* $(p_t^i)_{t=1}^T$ *is at most* $2T\sqrt{|\mathcal{A}|\tilde{\epsilon}}$

*Proof.* From lemma B.13 at each time step the difference of expected utility is bounded by $\sqrt{|\mathcal{A}|\tilde{\epsilon}}$ in absolute value. □

**Theorem B.19.** *If at time $t$ player $i$ plays according to* $\tilde{p}_t^i$, *where* $\tilde{\epsilon}=\frac{T^{-1/4}}{|\mathcal{A}|}$ *and* $p_t=\tilde{p}_t^1(\tilde{p}_t^2)^\top$, *then the regret for playing according to* $\tilde{p}_t^i$ *is at most* $O(T^{7/8})$. *Further* $\limsup_{T\to\infty}\left\|\tilde{\mathrm{M}}-\widehat{\mathrm{M}}\right\|_2=0$, *almost surely. Additionally if* $\tilde{\sigma}=\frac{1}{T}\sum_{t=1}^T \delta_t$ *is the stationary distribution of* $\tilde{\mathrm{M}}$ *corresponding to the observed play and* $\widehat{\sigma}=\frac{1}{T}\sum_{t=1}^T p_t$ *is the stationary distribution of* $\widehat{\mathrm{M}}$ *corresponding to the averaged empirical distribution, then* $\limsup_{T\to\infty}\left\|\tilde{\sigma}-\widehat{\sigma}\right\|_1=0$ *almost surely.*

*Proof.* Set $\tilde{\epsilon}=\frac{T^{-1/4}}{|\mathcal{A}|}$. The regret bound of the no-external regret algorithms now becomes $O(T^{7/8})$. We can, however, now guarantee that $\sum_{t=1}^T p_t(x_i)\ge\frac{T^{-1/4}}{|\mathcal{A}|}$ and thus, combining this with the high probability bound we obtain that with probability at least $1-C\exp\left(-\frac{T\epsilon^2}{4}\right)$ it holds that

$|\tilde{\mathrm{M}}_{i,j} - \widehat{\mathrm{M}}_{i,j}| < 2\epsilon \frac{T^{1/4}}{|\mathcal{A}|}$. To see this, let $x = \frac{1}{T}\sum_{t=1}^{T} p_{t-1}(x_i)p_t(x_j), \widehat{x} = \frac{1}{T}\sum_{t=1}^{T}\delta_{t-1}(x_i)\delta_t(x_j), y = \frac{1}{T}\sum_{t=1}^{T} p_t(x_i), \widehat{y} = \frac{1}{T}\sum_{t=1}^{T}\delta_t(x_i)$. Then

$$|\tilde{\mathrm{M}}_{i,j} - \widehat{\mathrm{M}}_{i,j}| = \left|\frac{x}{y} - \frac{\widehat{x}}{\widehat{y}}\right| \le \frac{|x - \widehat{x}|}{|y|} + \frac{|\widehat{x}|}{|1/y - 1/\widehat{y}|} \le \frac{\epsilon}{\widetilde{\epsilon}} + \frac{|\widehat{x}||y - \widehat{y}|}{|y\widehat{y}|} \le 2\frac{\epsilon}{\widetilde{\epsilon}},$$

where the last inequality holds because $\widehat{x} \le \widehat{y}$. Setting $\epsilon = T^{-1/3}$ and a union bound we arrive at $\mathbb{P}\left[\|\tilde{\mathrm{M}} - \widehat{\mathrm{M}}\|_2 > T^{-1/12}\right] < C\exp\left(-\frac{T^{1/3}}{4}\right)$. By Borel-Cantelli lemma we have $\limsup_{T\to\infty}\|\tilde{\mathrm{M}} - \widehat{\mathrm{M}}\|_2 = 0$ almost surely. From lemma B.16 and a union bound we know that with $\mathbb{P}\left[\|\tilde{\sigma} - \widehat{\sigma}\|_1 > \epsilon\right] < 2|\mathcal{A}|\exp\left(-\frac{T\epsilon^2}{4}\right)$. Setting $\epsilon = T^{-1/3}$ and again using Borel-Cantelli's lemma we see that $\limsup_{T\to\infty}\|\tilde{\sigma} - \widehat{\sigma}\|_1 = 0$. $\qquad\square$

# C    Extending the framework to arbitrary memories

**Definition C.1.** Let player 1 have memory $m_1$ and player 2 have memory $m_2$. Let the function spaces be $\mathcal{F}_1 = \{f : \mathcal{A}_2^{m_1} \to \mathcal{A}_1\}$ and $\mathcal{F}_2 = \{f : \mathcal{A}_1^{m_2} \to \mathcal{A}_2\}$. Let $\pi$ be a distribution over $\mathcal{F}_1 \times \mathcal{F}_2$. Define the $m$-memory bounded Markov process, where $m = \max(m_1, m_2)$ to be $\mathbb{P}\left[(a_t, b_t)|(a_{t-1}, b_{t-1}), \cdots, (a_{t-m}, b_{t-m})\right] = \sum_{(f,g)\in\mathcal{F}_1\times\mathcal{F}_2:f(b_{t-1},\cdots,b_{t-m_1})=a_t,g(a_{t-1},\cdots,a_{t-m_2})=b_t}\pi(f,g)$. We associate with this Markov process the matrix $\mathrm{M} \in \mathbb{R}^{|\mathcal{A}|^m\times|\mathcal{A}|^m}$, with $\mathrm{M}_{(x_{t-1},\cdots,x_{t-m}),(x_t,x_{t-1},\cdots,x_{t-m+1})} = \mathbb{P}\left[x_t|x_{t-1},\cdots,x_{t-m}\right]$ and $\mathrm{M}_{i,j} = 0$ for all other entries.

**Definition C.2.** The utility of $\pi$ is defined through the stationary distribution $\gamma$ of $\mathrm{M}$. In particular $\gamma$ is a distribution over $|\mathcal{A}|^m$ with entries indexed by $(x_t, \cdots, x_{t-m+1})$. Let $\sigma$ be the marginal distribution of $x_t$, then $u_i(\pi) = \sup_\sigma \mathbb{E}_{(a,b)\sim\sigma}\left[u_i(a,b)\right]$

**Definition C.3.** The empirical $m$-memory bounded Markov process is $\widehat{\mathrm{M}}$ with $\mathbb{P}\left[x_1|x_2,\cdots,\mathrm{x}_{m+1}\right] = \frac{\sum_t \prod_{i=0}^{m-1} p_{t-i}(x_{i+1})\times\prod_{i=0}^{m-1} p_{t-i}(x_{i+2})}{\sum_t \prod_{i=0}^{m-1} p_{t-i}(x_{i+2})}$

**Theorem C.4.** *The distribution $\widehat{\sigma}$ over $|\mathcal{A}|^m$ with entries indexed as*

$$\widehat{\sigma}((a_1, b_1), \cdots, (a_m, b_m)) = \frac{1}{T}\sum_t \prod_{i=0}^{m-1} p_{t-i}((a_{i+1}, b_{i+1}))$$

*, is a stationary distribution of $\widehat{\mathrm{M}}$.*

*Proof.* Consider $(\widehat{\sigma}^\top\widehat{\mathrm{M}})_{x_1,\cdots,x_m}$, we show it is equal to $\widehat{\sigma}(x_1,\cdots,x_m)$:

$$
\begin{aligned}
(\widehat{\sigma}^\top\widehat{\mathrm{M}})_{x_1,\cdots,x_m} &= \sum_{x_2,\cdots,x_{m+1}} \widehat{\mathrm{M}}_{(x_2,\cdots,x_{m+1}),(x_1,\cdots,x_m)}\widehat{\sigma}(x_2,\cdots,x_{m+1}) \\
&= \sum_{x_2,\cdots,x_{m+1}} \mathbb{P}\left[x_1|x_2,\cdots,x_{m+1}\right]\widehat{\sigma}(x_2,\cdots,x_{m+1}) \\
&= \frac{1}{T}\sum_{x_2,\cdots,x_{m+1}} \frac{\sum_t \prod_{i=0}^{m-1} p_{t-i}(x_{i+1})\times\prod_{i=0}^{m-1} p_{t-i}(x_{i+2})}{\sum_t \prod_{i=0}^{m-1} p_{t-i}(x_{i+2})}\sum_t \prod_{i=0}^{m-1} p_{t-i}(x_{m-i+2}) \\
&= \frac{1}{T}\sum_{x_2,\cdots,x_{m+1}}\sum_t \prod_{i=0}^{m-1} p_{t-i}(x_{i+1})\times\prod_{i=0}^{m-1} p_{t-i}(x_{i+2}) \\
&= \frac{1}{T}\sum_t \prod_{i=0}^{m-1} p_{t-i}(x_{i+1}) = \widehat{\sigma}(x_1,\cdots,x_m).
\end{aligned}
$$

$\qquad\square$

We also need a theorem which states that the utility $\mathbb{E}_{(f,g)\sim\pi}\left[u_1(a, g(a,\cdot,a)\right]$ is the expectation of the stationary distribution of the Markov process coming from the play $(a, g(a,\cdots,a))$ according to the marginal of $g$.

**Definition C.5.** Let $\mathrm{M}_1^a$ be the $m_2$-memory bounded Markov process which comes from player 1 playing a fixed action $a \in \mathcal{A}_1$ and player 2 playing $g \in \mathcal{F}_2$ according to the marginal of $\pi$, i.e. $(\mathrm{M}_1^a)_{(x_1,\cdot,x_{m_2}),(x_1,\cdot,x_{m_2+1})} = \mathbb{P}\left[x_{m_2+1}|x_1,\cdots,x_{m_2}\right] = \sum_{(f,g):g(a_1,\cdots,a_{m_2})=b_{m_2+1}} \pi(f,g)$ if $a_{m_2+1} = a$ or 0 otherwise (here $x_i = (a_i, b_i)$). Similarly let $\mathrm{M}_2^b$ be the $m_1$-memory bounded Markov process which arises when player 2 switches to playing the constant action.

**Theorem C.6.** *Consider $\mathrm{M}_1^a$ and let $m = m_2$. Let $\bar{\sigma}$ be the following distribution over $\mathcal{A}^m$ — $\bar{\sigma}(x_1,\cdots,x_m) = \sum_{(f,g):g(a,\cdot,a)=b} \pi(f,g)$ if $x_1 = \cdots = x_m = (a,b)$ and 0 otherwise. Then $\bar{\sigma}$ is a stationary distribution of $\mathrm{M}_1^a$ and its marginal distribution $\gamma(a,b) = \sum_{(f,g):g(a,\cdot,a)=b} \pi(f,g)$ is such that $\mathbb{E}_{(a,b)\sim\gamma}\left[u_1(a,b)\right] = \mathbb{E}_{(f,g)\sim\pi}\left[u_1(a,g(a,\cdots,a))\right]$.*

*Proof.*

$$
\begin{aligned}
(\bar{\sigma}^\top \mathrm{M}_1^a)_{x_2,\cdots,x_{m+1}} &= \sum_{x_1} \bar{\sigma}(x_1,\cdots,x_m)(\mathrm{M}_1^a)_{(x_1,\cdots,x_m),(x_2,\cdots,x_{m+1})} \\
&= \sum_{x_1} \bar{\sigma}(x_1,\cdots,x_m)\mathbb{P}\left[x_{m+1}|x_1,\cdots,x_m\right] \\
&= \sum_{b_1\in\mathcal{A}_2} \sigma((a,b_1),\cdots,(a,b_1)) \sum_{f,g(a,\cdots,a)=b_{m+1}} \pi(f,g) \\
&= \sum_{b_1\in\mathcal{A}_2}\left[\sum_{f,g(a,\cdots,a)=b_1}\pi(f,g)\right]\sum_{f,g(a,\cdots,a)=b_{m+1}}\pi(f,g) \\
&= \sum_{f,g(a,\cdots,a)=b_{m+1}}\pi(f,g) = \sigma((a,b_{m+1}),\cdots,(a,b_{m+1})) = \sigma(x_2,\cdots,x_{m+1})
\end{aligned}
$$

$\square$

The rest of the proofs extend in similar ways to the case with general memory. There is the question of not requiring $\bar{\sigma}$ to be a stationary distribution but to be any distribution such that the marginal with respect to the first coordinate of $\bar{\sigma}^\top\mathrm{M}_1^a$, satisfies the expectation equality $\mathbb{E}_{(a,b)\sim\gamma}\left[u_1(a,b)\right] = \mathbb{E}_{(f,g)\sim\pi}\left[u_1(a,g(a,\cdots,a))\right]$.

## D   Extending the framework to arbitrary number of players

**Definition D.1.** Consider an $n$ player game where player $i$ has memory $m_i$. Define the set of functions $\mathcal{F}_i = \{f : \mathcal{A}_{-i}^{m_i} \to \mathcal{A}_i\}$. We consider a distribution $\pi$ over $\mathcal{F} = \times_{i=1}^n \mathcal{F}_i$. Let $m = \max(m_1,\cdots,m_n)$. Let $x = (a^1,\cdots,a^n) \in \mathcal{A}$ and let $x^{-i} = (a^1,\cdots,a^{i-1},a^{i+1},\cdots,a^n) \in \mathcal{A}_{-i}$. Define the $m$-memory Markov process $\mathrm{M}$ such that $\mathbb{P}\left[x_{m+1}|x_1,\cdots,x_m\right] = \sum_{(f_1(x_m^{-1},\cdots,x_{m-m_1+1}^{-1})=a_{m+1}^1,\cdots,f_n(x_m^{-n},\cdots,x_{m-m_n+1}^{-n})=a_{m+1}^n)} \pi(f_1,\cdots,f_n)$.

All other definitions follow the same form. The problem with this most general setting is – how do we construct no-policy regret algorithms and what does it even mean to have no-policy regret? The utility function of player $i$ at time $t - u_i(\cdot, x_t^{-i})$ is no longer interpretable is an $m$-memory bounded function, since $x_t^{-i}$ depends on all other players' memories. One possible solution is to look at this utility as an $m$-memory bounded function, where $m$ is the maximum memory among all other players.

## E   Simple example of a policy equilibrium

We now present a simple 2-player game with strategies of the players which lead to a policy equilibrium, which in fact is not a CCE. Further these strategies give the asymptotically maximum utility for both row and column players over repeated play of the game. The idea behind the construction is very similar to the one showing incompatibility of policy regret and external regret. The utility matrix for the game is given in Table 2. Since the column player has the same payoff for all his actions they will always have no policy and no external regret. The strategy the column player chooses is to always play the function $f : \mathcal{A}_1 \to \mathcal{A}_2$:

$$
f(x) = \begin{cases} c & x = a \\ d & x = b. \end{cases}
$$

Table 2: Utility matrix

| Player 1\Player 2 | c | d |
|---|---|---|
| **a** | (3/4,1) | (0,1) |
| **b** | (1,1) | (0,1) |

In the view of the row player, this strategy corresponds to playing against an adversary which plays the familiar utility functions:

$$u_t(a_{t-1}, a_t) = \begin{cases} 1 & a_{t-1} = a, a_t = b \\ \frac{3}{4} & a_{t-1} = a_t = a \\ 0 & \text{otherwise.} \end{cases}$$

We have already observed that on these utilities, the row player can have either no policy regret or no external regret but not both. What is more the utility of no policy regret play is higher than the utility of any of the no external regret strategies. This already implies that the row player is better off playing according to the no policy regret strategy which consists of always playing the fixed function $g : \mathcal{A}_2 \rightarrow \mathcal{A}_1$ given by $g(x) = a$. Below we present the policy equilibrium $\pi \in \Delta \mathcal{F}$, corresponding Markov chain $\mathrm{M} \in \mathbb{R}^{|\mathcal{A}| \times |\mathcal{A}|}$ and its stationary distribution $\sigma \in \Delta \mathcal{A}$ satisfying the no policy regret requirement.

$$\pi(\tilde{f}, \tilde{g}) = \delta_{(f,g)}, \mathrm{M} = \begin{matrix} & \begin{matrix} (a,c) & (a,d) & (b,c) & (b,d) \end{matrix} \\ \begin{matrix} (a,c) \\ (a,d) \\ (b,c) \\ (b,d) \end{matrix} & \begin{pmatrix} 1 & 0 & 0 & 0 \\ 1 & 0 & 0 & 0 \\ 0 & 1 & 0 & 0 \\ 0 & 1 & 0 & 0 \end{pmatrix} \end{matrix}, \sigma(x,y) = \delta_{(a,c)}.$$

Suppose the row player was playing any no policy regret strategy, for example one coming from a no policy regret algorithm, as a response to the observed utilities $u_t(\cdot, \cdot)$. Since the only sublinear policy regret play for these utilities is to only deviate from playing $a$ a sublinear number of times we see that the empirical distribution of play for the row player converges to the dirac distribution $\delta_a$. Together with the strategy of the column player, this implies the column player chooses the action $d$ only a sublinear number of times and thus their empirical distribution of play converges to $\delta_c$. It now follows that the empirical distribution of play converges to $\delta_a \times \delta_c = \delta_{(a,c)} \in \Delta \mathcal{A}$. We can similarly verify that the empirical Markov chain will converge to $\mathrm{M}$ and the empirical functional distribution $\widehat{\pi}$ converges to $\pi$. Theorem B.9 guarantees that because both players incur only sublinear regret $\pi$ is a policy equilibrium. It should also be intuitively clear why this is the case without the theorem – suppose that the row player switches to playing the fixed action $b$. The resulting functional distribution, Markov chain and stationary distributions become:

$$\pi_b(\tilde{f}, \tilde{g}) = \delta_{(f, \widehat{g} \equiv b)}, \mathrm{M}_b = \begin{matrix} & \begin{matrix} (a,c) & (a,d) & (b,c) & (b,d) \end{matrix} \\ \begin{matrix} (a,c) \\ (a,d) \\ (b,c) \\ (b,d) \end{matrix} & \begin{pmatrix} 0 & 0 & 1 & 0 \\ 0 & 0 & 1 & 0 \\ 0 & 0 & 0 & 1 \\ 0 & 0 & 0 & 1 \end{pmatrix} \end{matrix}, \sigma_b(x,y) = \delta_{(b,d)}.$$

The resulting utility for the row player is now $0$, compared to the utility gained from playing according to $\pi$, which is $3/4$.