[Reviews · NeurIPS 2018]

Reviewer 1



This paper contributes a number of theoretical results regarding policy regret (previously introduced in Arora et al. 2012), and introduces and analyzes a new equilibrium concept called policy equilibrium. Specifically, the authors investigate the relationship between external regret and policy regret, and demonstrate that sublinear regret in one does not imply sublinear regret in the other. Next, the authors introduce the concept of a policy equilibrium, and demonstrate that coarse correlated equilibria are a subset of policy equilibria. The paper is purely theoretical and does not present any experimental results. The contributions of this paper seem to me to be significant and surprising. In particular, the fact that low external regret does not imply low policy regret, and vice versa, but that nevertheless coarse correlated equilibria are a subset of policy equilibria is an interesting result. The theoretical analysis is also highly non-trivial and seems to me to merit publication. The paper is well-written. As someone who is not an expert in bandit algorithms, I was able to follow both the intuition and the mathematics reasonably well, even though the paper is highly theoretical. I had some trouble around the definition of a stable no-regret algorithm, and in section 4.1. The paper would benefit from more intuition in those sections. One small suggestion I have is that since the authors are only minimizing policy regret for a fixed memory length m, it seems to me that the term for policy regret P(T,a) should instead be something like P_m(T,a) or P^m(T,a). Also, I think the paper would benefit from a greater discussion of the significance and applications of this work. The authors make a convincing argument that when dealing with adaptive adversaries, policy regret seems more appropriate than external regret. But external regret minimization is used in all sorts of practical applications, including some with adaptive adversaries. Are there concrete situations where policy regret minimization and policy equilibria would be more appropriate and lead to better results (for some definition of "better")?

Reviewer 2



The paper studies the notion of policy-regret, introduced by Arora et al (2012), in the context of repeated 2-player games. Policy regret is an adaptation of the standard external regret that captures counterfactual interactions between the player and the adversary in an online learning setting, but so far has not been seriously studied in the natural setting of repeated interaction between two players/agents in a game, interested in maximizing their own utilities. The paper addresses various different aspects of policy regret minimization in repeated games. Most notably, the authors give a precise characterization of the set of equilibria (termed “policy equilibria”) approached by two players in a repeated game when both follow policy-regret minimization algorithms, and show that this set strictly contains the set of all coarse-correlated equilibria (which are approached by classical external regret-minimizing algorithms). The authors’ study of the topic is thorough and the paper feels complete. The questions considered are natural and compelling, and the results established are creative, elegant and non-trivial. The writing is perhaps a bit too verbose but overall excellent. The main text feels polished and the technical proofs are clear and detailed. My only complaint (besides a couple of minor comments listed below) is that, as the precise definitions of “policy equilibria” and related concepts turn out to be quite complex, it would have been helpful to include few simple examples where these take a more explicit form and are easier to interpret. (This could be addressed in a future, longer version of the paper.) Overall, an excellent paper that has been a pleasure to read. I strongly support acceptance. Few minor comments: * Some corrections/additions to related work: the MWU and Exp3 algorithms are mentioned without a proper citation; the tight policy-regret bound for switching costs adversaries was in fact proved by Dekel, Ding, Koren & Peres (STOC’14) (and was later extended to more general movement costs by Koren, Livni & Mansour (COLT’17, NIPS’17)); another paper perhaps worths mentioning is Even-Dar, Mansour & Nadav (STOC’09) that studies regret minimization dynamics in concave games. * Theorem 3.2: the theorem statement is for m ≥ 2 but the proof (in the supplementary) seems to apply only for m ≥ 2 ...? Also, the quantifier “for any constant m ≥ 2” should be moved to the beginning of the theorem statement. * Section 4: I couldn’t find the definition of the action set \mathcal{A}. * For completeness, it would be worthwhile to include the definition of the Prokhorov metric and state Prokhorov’s theorem.

Reviewer 3



The paper proposes a definition of bounded-memory policy regret, then defines a corresponding equilibrium concept for two-player games, and proves results on (i) convergence (in a non-standard sense) of no-policy-regret strategies to policy equilibria, and (ii) relating policy equilibria to coarse correlated equilibria. The ideas presented in the paper are clearly motivated: a notion of regret which captures adaptive reaction of the adversary is useful to study. The technical results are also interesting in the context of learning in games. I found the paper hard to read, however, due to presentation issues. The definitions are often preceded by ambiguous statements. For example: - In the paragraph "Algorithm of the player" line 180: Alg_t is defined as a Alg_t: (A_1 \times A_2)^t \to \Delta A_1, but then redefined as a function from A_2^t \to \Delta A_1 in the full information case, and finally as a distribution over such functions in the bandit setting. In Definition 3.3, the second definition seems to be used. In the same definition, "a possibly random algorithm" is not clearly defined, since the randomness can refer either to randomization over the action set, or to randomization over functions f_t. Line 194: what is meant by "simulating the play of players"? It is possible the reader could guess a definition of \mu, but it would be much better to give a rigorous and unambiguous definition (e.g. in terms of a distribution of an appropriate Markov chain). This entire section should be replaced with an unambiguous mathematical description of the game setting. - In Section 4, the discussion leading to the main definition (Definition 4.5) often involves imprecise statements, e.g. "under reasonable play from all parties" (line 229), "The player believes their opponent might be" (line 231). \epsilon is undefined on line 237. - When switching between distributions over the action set A and the joint policy set F, the presentation can be improved. I believe it would be better to start by defining the equilibrium concept (Definition 4.5) then discuss its interpretation. - \tilde \sigma_a and \tilde \sigma_b (line 336) are not clearly defined. Besides presentation, several points require discussion and justification: 1) Modifying the definition of policy regret (Definition 3.1): overloading existing definitions has serious drawbacks. In this case it is even harder to justify, since the whole point of defining policy regret is to take into account the adaptivity of the adversary (a point that the paper highlighted several times). The sequence of plays in the proposed definition simply does not reflect a fixed action policy. 2) Justification of the new equilibrium concept (policy equilibrium). Policy equilibrium compares stationary distribution of Markov chains. This needs careful justification and interpretation. For example, do the players care about their stationary distributions because the game is played infinitely often? Why is this a better equilibrium concept than existing concepts for repeated games? How does it generalize to n players? The proposed concept only applies in a context of repeated play (unlike other concepts mentioned in the paper, e.g., Nash equilibria and correlated equilibria). It should be related and compared to other equilibrium concepts of repeated games. 3) The non-uniquess of stationary distributions also raises some issues (the ability of the players to select these distributions). The comment following Definition 4.5 ignores the issue by referring to "the stationary distribution", omitting that there may be more than one. ================= Thank you for your responses and clarifications, in particular regarding Definition 3.1, this addresses one of my concerns. I agree that Definition 4.5 does not rely on the uniqueness of the stationary distribution, but the point is rather the motivation of this new equilibrium concept: when there are multiple stationary distributions, the play could converge to a different stationary distribution than the one satisfying the guarantees, and this is problematic. As raised during discussion with other reviewers, a better justification could be that one can ensure uniqueness of the stationary distribution at a relatively small cost, by mixing in a uniform distribution. A more careful discussion should be included in the revision.